# OPEN-WORLD
# PEDESTRIAN TRAJECTORY PREDICTION

## ABSTRACT

Most deep learning-based pedestrian trajectory prediction models are trained offline, which significantly limits their performance when encountering novel motion patterns in open-world environments. To endow trajectory prediction agents with lifelong learning, we introduce the Open-World Pedestrian Trajectory Prediction (OWPTP). OWPTP requires models to autonomously detect distribution shifts in motion patterns, continually accommodate novel pattern information, and retain previously acquired knowledge. However, motion patterns are abstract and ill-defined. Our analysis indicates that the dominant source of motion pattern discrimination arises from trajectory epistemic uncertainty tied to pedestrian goals. Based on this insight, we propose Goal-based Motion Pattern Detection and Replay (GMPDR) framework. By modeling epistemic uncertainty, GMPDR extracts pattern-related trajectory features and builds an explicit instance-to-pattern mapping through dual contrast modules to delineate motion pattern boundaries. On top of this mapping, we formulate hyperspherical novelty detection and sparse, representative replay mechanisms at the motion-pattern level. These mechanisms respectively achieve novelty detection anchored to model-defined patterns and accommodation that preserves the semantic integrity of the patterns. The framework is extensible and integrates seamlessly with various existing trajectory predictors. Experiments demonstrate that GMPDR effectively adapts to novelty and reduces forgetting. The anonymous code link is provided in the reproducibility statement.

## 1 INTRODUCTION

Pedestrian trajectory prediction aims to forecast the movements of multiple agents based on their past trajectories (Xu et al., 2022). Due to the complexity of pedestrian motion, trajectory prediction models deployed in open-world environments often encounter unfamiliar motion patterns (Wu et al., 2022). However, deep learning-based models typically depend on offline training (Habibi et al., 2020). Their generalization capabilities are challenged when novelty emerges, as the distribution of training data may differ from that of the test samples (Knoedler et al., 2022). To operate effectively, models must continually detect and accommodate novel motion patterns (Fig. 1(a)) (Gummadi et al., 2022). We refer to this crucial paradigm as Open-World Pedestrian Trajectory Prediction (OWPTP).

OWPTP can be decomposed into two phases: detecting novel motion patterns and accommodating new knowledge, which can be framed as out-of-distribution (OOD) detection and continual learning (CL). OOD or novelty detection aims to determine whether a given trajectory sample belongs to known in-distribution (ID) motion patterns or unknown OOD patterns (Nguyen et al., 2015). Upon detecting novelty, CL aims to adapt the model to new knowledge while effectively mitigating catastrophic forgetting that arises from parameter overwriting (Fig. 1 (b)) (McCloskey & Cohen, 1989).

OWPTP emphasizes detection and accommodation at the motion-pattern level. Existing trajectory-related methods fail to handle both phases simultaneously and are restricted to the instance level. Current trajectory OOD detection approaches typically identify anomalies by quantifying deviations between predicted and actual trajectories (Noghre et al., 2024). However, such methods can only assess whether an individual trajectory is anomalous and rely on future actual trajectories. Research on continual trajectory prediction often employs replay mechanisms, preserving a subset of known trajectories and integrating them into subsequent training to mitigate forgetting (Wu et al., 2022). Nevertheless, these methods incur high memory costs to support replay across diverse patterns.

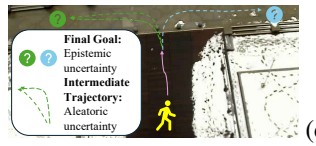

Figure 1: (a) The model fails in predicting novel motion. (b) Model accommodating novel knowledge leads to forgetting. (c) Epistemic uncertainty reflects unknown in the pedestrian's final goals.

Motion patterns are inherently abstract and ill-defined concepts, whereas trajectory instances represent concrete observations. To achieve learning at the motion-pattern level, OWPTP necessitates an explicit mapping between instances and patterns. Prior research indicates that features encoded from trajectories capture their pattern information (Fragkedaki et al., 2024). However, this information remains implicit and inadequate. To better focus on the essential components of motion patterns and extract discriminative features, we model trajectory patterns by decomposing trajectory uncertainty into epistemic and aleatoric components (Fig. 1(c)) (Mangalam et al., 2021). Epistemic uncertainty pertains to sources that are known to the agent but unknown to the model. It can be primarily understood as the model's insufficient knowledge regarding the pedestrian's intended goal, corresponding to the question of determining the destinations. Aleatoric uncertainty refers to uncertainty sources that are unknown to both the agent and the model. It encompasses the randomness in decisions and intents of other agents, which influence how pedestrians reach their goals (Kahneman, 2011).

We argue that the essence of assessing the novelty and distinctiveness of motion patterns stems partly from epistemic uncertainty. This is because novel motion patterns largely manifest as goal shifts: people head to different destinations under changing contexts, whereas responses to environmental and social cues are comparatively regular. Recent works have demonstrated that numerous methods first predict goals and subsequently employ the predicted goals to assist in intermediate trajectory refinement, corresponding to the goal prediction and trajectory refinement stages (Xu et al., 2022; Yue et al., 2022; Lin et al., 2024a). Our experiments demonstrate that, when a series of patterns is learned, approximately **85%** of the forgetting occurs in the goal prediction module. Therefore, we highlight that a goal-based framework offers a promising approach to realizing OWPTP.

This paper proposes a Goal-based Motion Pattern Detection and Replay (GMPDR) framework. By estimating the probability distribution of goals, the framework constructs an encoder-decoder goal prediction model to capture epistemic uncertainty (Mangalam et al., 2021). Based on this, GMPDR further enhances its capacity to extract motion pattern-specific features through branch expansion. To establish the mapping between trajectory instances and motion patterns, we introduce a theoretically grounded Motion Pattern Dual Contrast (MPDC) module. MPDC first clusters trajectory features into motion patterns through set-level contrastive learning (Li et al., 2021). Subsequently, it uses the clustering results as priors to construct hyperspherical embedding space via instance-level contrastive learning. Within this space, GMPDR autonomously defines the boundaries of known motion patterns, enabling hyperspherical OOD detection. To alleviate forgetting during accommodation, GMPDR selects trajectories based on the mapping relationships, constructing sparse, representative replay that preserve semantic diversity across patterns. Our contributions are threefold:

- To address the challenge of complex and dynamic pedestrian trajectory prediction in open-world environments, we introduce the OWPTP paradigm. Empirical and experimental analysis demonstrates that continual trajectory forecasting is bottlenecked by the goal prediction stage.

- We propose GMPDR, which employs a Motion Pattern Dual Contrast module grounded in theory. This MPDC module achieves hyperspherical OOD detection and representative sparse replay at the motion-pattern level, corresponding to autonomous detection and accommodation functions.

- GMPDR is a goal prediction framework that can be integrated with various trajectory refinement modules. This enables the extension of existing trajectory prediction algorithms into OWPTP.

## 2 RELATED WORK

**Pedestrian trajectory prediction.** Deep learning-based trajectory prediction approaches typically focus on leveraging scene information or extracting features related to motion and interactions (Lin et al., 2024a; Yue et al., 2022; Xiang et al., 2024). Numerous methods can be interpreted as a two-stage process: goal prediction followed by trajectory refinement. For instance, YNet proposes a

mutually coupled goal and trajectory decoder, which employs skip connections to effectively embed goal information (Mangalam et al., 2021). ExpertNet constructs a query-based framework that retrieves detailed trajectory goals to guide trajectory refinement (Zhao & Wildes, 2021). MemoNet introduces a dual-memory structure to store representative instances (Xu et al., 2022). Our analysis ultimately reveals that the goal prediction is crucial to achieving OWPTP.

**Open-world Learning.** Models deployed in open-world environments require detection and accommodation of novelty, which correspond to OOD detection and CL, respectively. However, most research focuses only on either OOD detection or CL individually (Kim et al., 2025). Replay-based CL methods have been widely explored, replaying previous samples to mitigate forgetting (Hayes et al., 2020; Tong et al., 2023; Guo et al., 2022; Wang et al., 2023; Yan et al., 2021; Wang et al., 2022; Hu et al., 2023). Meanwhile, OOD detection highlights the necessity for the capability to reject OOD instances (Morteza & Li, 2022; Tao et al., 2023; Jiang et al., 2024; Sun et al., 2022; Du et al., 2022; Sehwag et al., 2021; Guille-Escuret et al., 2023). Few studies have addressed detection and accommodation jointly, often employing multiple expert networks, termed task-agnostic methods (Zhu et al., 2024; Zeno et al., 2021; Lee et al., 2020). However, such approaches fail to decouple the processes into OOD detection and CL, leading to suboptimal performance. SHELS (Gummadi et al., 2022) is a rare method that explicitly separates detection and accommodation, achieving this through orthogonal embedding and regularization. Furthermore, these prior methods often remain limited to supervised image classification tasks and struggle to generalize to complex scenarios.

Exploring OWPTP enables models to adapt to evolving motion patterns. Works most relevant to OWPTP have focused exclusively on either accommodating new trajectories or detecting abnormal trajectories. Moreover, these approaches either rely on manually defined task boundaries for accommodation (Wu et al., 2022), or depend on future actual trajectories for detection (Noghre et al., 2024). To our knowledge, no existing research has specifically addressed the OWPTP problem.

## 3 FRAMEWORK ANALYSIS

### 3.1 OWPTP FORMALIZATION

Let $\mathcal{M}$ denote a trajectory predictor deployed in open-world environments. The objective of $\mathcal{M}$ is to predict the future trajectory $\mathbf{R}_{\text{pred}}^{(i)} = \left[\mathbf{r}_{w_{obs}+1}^{(i)}, \ldots, \mathbf{r}_{w_{obs}+w_{fut}}^{(i)}\right]$ based on the observation $\mathbf{R}_{\text{obs}}^{(i)} = \left[\mathbf{r}_0^{(i)}, \ldots, \mathbf{r}_{w_{obs}}^{(i)}\right]$ of pedestrian $i$, along with other complementary information. Here, $\mathbf{r}$ denotes the 2D coordinates, $w_{obs}, w_{fut}$ represent the lengths of the observation and prediction window. We assume that model $\mathcal{M}$ has been trained on dataset $D_{0:C}$, and there are $C$ abstract motion patterns within the data domain $\mathcal{X}_{0:C}^{ID}$. During inference, $\mathcal{M}$ performs detection and prediction. If the input $\mathbf{x}_i$ matches a known pattern $c$, the model predicts normally. However, when $\mathbf{x}_i$ belongs to an unknown motion pattern, $\mathcal{M}$ identifies $\mathbf{x}_i$ as an OOD sample and accumulates this detection result. Based on these detections, $\mathcal{M}$ autonomously transitions to a training phase upon detecting the emergence of a novel motion pattern. Subsequently, $\mathcal{M}$ conducts accommodation using dataset $D_{C+1}$ associated with the new pattern, updating the known data domain to $\mathcal{X}_{0:C+1}^{ID}$, as Fig. 2(a). It is important to note that multiple novel motion patterns may be detected and accommodated simultaneously.

### 3.2 GOAL-BASED FRAMEWORK

By analyzing the essence of motion patterns, we identify the key aspects for OOD detection as well as the specific locations where forgetting occurs. It has been noted that uncertainty in pedestrian trajectories can be decomposed into epistemic and aleatoric components, corresponding to the goal and intermediate trajectory. When accounting for environments, an intended goal reflects a unique pattern, whereas the trajectory refinement strategy is shared across patterns. This means that epistemic uncertainty and its carrier, the goals, often correspond to the distinctiveness of motion patterns.

Prior work has also shown that jointly encoding goals enhances algorithmic performance (Zhao & Wildes, 2021). We assume that $\mathcal{M}$ can be decoupled into two sub-modules: goal prediction $\mathcal{M}_{goal}$ and trajectory refinement $\mathcal{M}_{traj}$. Given observed trajectory $\mathbf{R}_{\text{obs}}^{(i)}$ of pedestrian $i$, the associated complementary information $S_{goal}^{(i)}$ and $S_{traj}^{(i)}$ required by the two sub-modules, the output is:

$$\mathbf{R}_{\text{pred}}^{(i)} = \mathcal{M}_{traj}(\mathbf{R}_{\text{obs}}^{(i)}, S_{traj}^{(i)}, \mathbf{G}_{\text{pred}}^{(i)}), \quad \mathbf{G}_{\text{pred}}^{(i)} = \mathcal{M}_{goal}(\mathbf{R}_{\text{obs}}^{(i)}, S_{goal}^{(i)}), \tag{1}$$

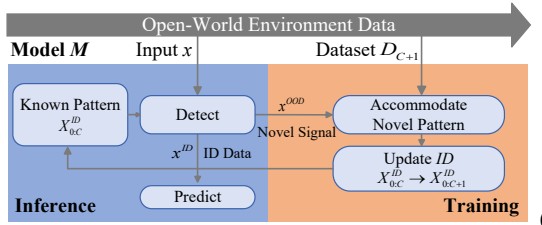
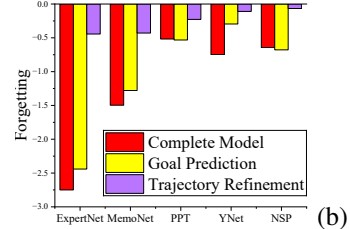

Figure 2: (a) OWPTP paradigm with two phases. (b) Average Displacement Error forgetting.

where $\mathbf{G}_{\text{pred}}^{(i)}$ denotes the predicted goal. We validate across different methods, focusing on which module is more sensitive to forgetting. This is achieved by freezing the module weights during training. As shown in Fig. 2(b), all methods exhibit some degree of performance degradation after continual accommodation. However, the goal prediction module experiences significant forgetting compared to the trajectory refinement module. Therefore, we argue that addressing OWPTP should prioritize improvements in the goal prediction module, termed the goal-based framework.

While replay is effective in preventing forgetting, the conventional replay necessitates a complete set of trajectories captured simultaneously. This is because the refinement module relies on the social interaction context $S_{traj}^{(i)}$. However, $S_{traj}^{(i)}$ is redundant for the goal-based framework, therefore only sparse samples are needed to fulfill the replay function. Moreover, the goal-based framework allows integration with various trajectory refinement modules, which can be either frozen or fine-tuned to maintain the performance. We present further analysis of the goal-based framework in Appendix A.

## 4 GOAL-BASED MOTION PATTERN DETECTION AND REPLAY

The proposed GMPDR is a goal-based framework. Its core is to enhance focus on motion-pattern discriminability by learning epistemic uncertainty, thereby achieving OOD detection and representative replay at the motion-pattern level. Unlike classical methods that rely directly on supervised information to construct memory and define distribution boundaries, GMPDR establishes an unsupervised mapping between trajectories and motion patterns. Specifically, GMPDR employs an encoder-decoder architecture, where trajectory mapping is carried out by MPDC modules. Upon detecting novelty, GMPDR triggers an accommodation process. First, it updates the predictor using novel data along with previous replay samples. Subsequently, the corresponding MPDC module is optimized to incorporate the novel patterns into the ID space and select new representative replay samples. Once the accommodation process is complete, the system resumes the inference phase to carry out subsequent detection and prediction tasks. The pipeline is illustrated in Fig. 3.

### 4.1 CONTINUAL GOAL PROBABILITY DISTRIBUTION PREDICTION

GMPDR first addresses goal prediction, which employs a U-Net encoder $U_e$, a goal decoder $U_g$, and an auxiliary trajectory decoder $U_t$. For input $\mathbf{x}_i$, GMPDR only requires the observation $\mathbf{R}_{\text{obs}}^{(i)}$ and the complementary information $S_{goal}^{(i)}$, which refers to a scene segmentation map. To align inputs, we represent observations as heatmaps, where the values decay with distance from observed points. For OWPTP, GMPDR must not only complete the final goal prediction, but also needs to extract motion-pattern-related trajectory features by encoder $U_e$. These features are transmitted through skip connections to the goal decoder $U_g$, which generates the goal probability distribution map. The learned goal probability reflects the epistemic uncertainty in trajectories, thereby compelling $U_e$ to encode trajectory and scene information into features that represent corresponding patterns (Fragkedaki et al., 2024; Deng et al., 2024). Additionally, the trajectory decoder $U_t$ utilizes intermediate trajectory information to stabilize and facilitate the learning process.

In the OWPTP process, GMPDR first conducts an initial training phase to optimize all parameters, enabling the model to develop cognitive and predictive capabilities. GMPDR freezes the parameters of $U_e$ after the initial training. This fixed parameter space prevents shifts in the encoding of previously acquired patterns for detection. Meanwhile, to accommodate new knowledge, GMPDR allows the goal decoder $U_g$ to be updated and incorporates sparse replay to alleviate forgetting. Inspired by prior work (Mangalam et al., 2021), the encoder-decoder architecture incorporates both convolutional and deconvolutional blocks. The details of the goal predictor are provided in Appendix B.

Figure 3: Overview of GMPDR, which includes a goal prediction model and MPDC modules. Whenever confronted with a novel pattern set, MPDC first projects features onto clustering bases. Subsequently, the clustering guides each embedding head to filter the ID and OOD samples.

## 4.2 MOTION PATTERN DUAL CONTRAST MODULE (MPDC)

We propose abstracting trajectories into motion patterns, and thus patterns can serve as units for detection and replay selection. Suppose there exists a trajectory dataset $X$ that can be abstracted into $N$ patterns. We define this pattern set $M$ as task $T$. MPDC employs contrastive learning to perform clustering of the samples $\mathbf{x}_i$ into $M$, and conducts OOD detection using $M$ as ID. To implement contrastive learning, the inputs are augmented to generate $X^a$ and $X^b$. Given that trajectory data encompass temporal and spatial information, we apply masking, rotation, and flipping augmentation.

Although features encoded during goal prediction are closely associated with motion patterns, the relationship captured by $U_e$ remains implicit. Therefore, MPDC introduces Low-Rank Adaptation (LoRA) (Hu et al., 2022) for each task, enabling the model to explicitly adapt features and enhance their discriminability. These weights are inserted into each block of $U_e$, and are optimized jointly with the downstream components. After feature extraction and adjustment via task-specific LoRAs, MPDC produces feature sets $F^a$ and $F^b$ that are suited for motion pattern clustering and embedding.

### 4.2.1 CLUSTERING CONTRASTIVE COMPONENT

For task $T$ with $N$ motion patterns, there exists a corresponding set of bases that serve as clusters. We assume that a specific feature $\mathbf{f}_i$ corresponds to a pattern category $n$ and an associated basis $\boldsymbol{\mu}_n$, where $n \in \{0, 1, ..., N\}$. By mapping the feature to all bases through an MLP projector $f_\theta$, we can then compute the probability that $\mathbf{f}_i$ belongs to cluster $n$ using the softmax function:

$$q_\theta(n \mid \mathbf{f}_i) = \exp\left(\boldsymbol{\mu}_n^\top f_\theta(\mathbf{f}_i)\right) / \sum_{j=1}^{N} \exp\left(\boldsymbol{\mu}_j^\top f_\theta(\mathbf{f}_i)\right). \tag{2}$$

To facilitate clustering, a clear separation between clusters is required. Building on set representation (Zaheer et al., 2017), MPDC represents clusters by constructing probability distributions based on the likelihood within a feature set. Specifically, in the clustering component, the output $\mathbf{y}_i \in \mathbb{R}^{N \times 1}$ of a single feature $\mathbf{f}_i$ is obtained by concatenating all $q_\theta(n \mid \mathbf{f}_i)$ in Eq. 2 along the category dimension. The overall output $Y \in \mathbb{R}^{N \times B}$ for the feature set $F$ is obtained by stacking all individual $\mathbf{y}_i$ along the batch dimension. Therefore, each column vector along the batch dimension $B$ represents the result of querying each cluster basis using the feature set (Shen et al., 2021).

We denote $\mathbf{p}_n$ as the set representation of the $n$th cluster basis. Subsequently, MPDC aggregates and separates various representations through contrastive learning. The augmented representations $\mathbf{p}_n^a$ and $\mathbf{p}_n^b$ are considered positive samples, whereas representations from different clusters are considered as negative samples. We adopt the SimCLR framework (Chen et al., 2020) to minimize:

$$\mathcal{L}_{clu} = -\frac{1}{N} \sum_{n=1}^{N} \log \frac{\exp(\mathbf{p}_n^{a\top} \mathbf{p}_n^b / \tau)}{\exp(\mathbf{p}_n^{a\top} \mathbf{p}_n^b / \tau) + \sum_{j=1}^{N} \exp(\mathbf{p}_j^\top \mathbf{p}_n / \tau) \, \mathbf{1}(j \neq n)}, \tag{3}$$

where $\mathbf{1}$ denotes the indicator function, and $\tau$ refers to the temperature. Ultimately, Eq. 3 explicitly correlates trajectories with motion patterns, establishing an unsupervised motion pattern clustering space. The assumption underlying this clustering is that cluster bases exist and are distinct. According to Lemma 1, contrastive loss leads to an asymptotically uniform embedding distribution. The objective of Eq. 3 implies that the set representations of different clustering categories should be distributed as evenly as possible, thereby increasing diversity among bases to satisfy the assumption.

### 4.2.2 EMBEDDING CONTRASTIVE COMPONENT

In the embedding component, we implement OOD detection by constructing multi-head hyperspherical embeddings for each motion pattern set. To enable effective separation between ID and OOD embeddings, we leverage the clustering probability $q_\theta(n \mid \mathbf{f}_i)$ to facilitate the transfer of pattern category. Assume that $I$ is the total sample size. For feature $\mathbf{f}_i$, we formulate its likelihood of being distinguishable from the other samples as $p_\theta(i \mid \mathbf{f}_i)$ through contrastive learning. We can derive the ELBO of $p_\theta(i \mid \mathbf{f}_i)$ with Jensen's inequality and KL divergence. See Appendix D for the proof:

$$\log p_\theta(i \mid \mathbf{f}_i) \geq \mathbb{E}_{q_\theta(n|\mathbf{f}_i)} \left[\log p_\theta(i \mid \mathbf{f}_i, n)\right] - \mathrm{KL}\left(q_\theta(n \mid \mathbf{f}_i) \,\|\, p_\theta(n \mid \mathbf{f}_i)\right). \tag{4}$$

For the second term, the true distribution $p_\theta(n \mid \mathbf{f}_i)$ of abstract motion patterns is unknown. Following prior work (Shen et al., 2021), we use a fixed uniform prior instead. Consequently, this term simplifies to an information entropy $\mathcal{H}$ form, $-\mathrm{KL}\left(q_\theta(n \mid \mathbf{f}_i) \,\|\, p_\theta(n \mid \mathbf{f}_i)\right) = \log N + \mathcal{H}\left(q_\theta(n \mid \mathbf{f}_i)\right)$. This formulation encourages a balanced distribution of trajectories across motion pattern categories.

The first term in Eq. 4 indicates that categorical priors are incorporated via conditional probabilities. Based on this, the embedding component assigns $N$ embedding heads, corresponding to the number of patterns. In the embedding space generated by the $n$th head with normalization $f_\theta^n$, features classified as pattern $n$ are treated as ID, whereas all other samples are regarded as OOD. Concurrently, each embedding head constructs a unit hyperspherical embedding space and a prototype. Every head aims to embed ID samples close to the prototype while pushing OOD samples away. This embedding space is modeled using the von Mises-Fisher (vMF) distribution (Mardia & Jupp, 2009),

$$p_d(\mathbf{a}; \mathbf{z}, \kappa) = Z_d(\kappa) \exp\left(\kappa \mathbf{a}^\top \mathbf{z}\right), \tag{5}$$

where $\kappa$ denotes the concentration parameter, and $Z_d(\kappa)$ is the normalization factor. By feeding all features into each head, we construct distinct hyperspherical embedding spaces for each motion pattern. Within each hyperspherical embedding space, ID samples associated with the corresponding motion pattern are pulled as close as possible to their respective prototype vector $\mathbf{z}_n$, thereby forming a well-defined decision boundary that separates them from OOD samples belonging to other patterns. The normalized probability of the optimization objective is formulated as:

$$\mathcal{L}_{emb} = -\frac{1}{NI} \sum_{n=1}^{N} \sum_{i=1}^{I} \log \frac{\exp(f_\theta^n(\mathbf{f}_i)^\top \mathbf{z}_n/\tau)\, \mathbf{1}(\hat{y}_i = n)}{\exp(f_\theta^n(\mathbf{f}_i)^\top \mathbf{z}_n/\tau) + \sum_{j=1}^{I} \exp(f_\theta^n(\mathbf{f}_j)^\top \mathbf{z}_n/\tau)\, \mathbf{1}(\hat{y}_j \neq n)}, \tag{6}$$

where $\hat{y}_i = \arg\max_{n \in \{0,1,...,N\}} q_\theta(n \mid \mathbf{f}_i)$. However, Eq. 6 relies entirely on the categorical priors provided by the clustering component, which may lead to unstable optimization during the early stages. Therefore, we further introduce an auxiliary contrastive loss, which directly aggregates positive sample pairs obtained via the auxiliary head $f_\theta^{aux}$ mapping. Let $\mathbf{q}_i = f_\theta^{aux}(\mathbf{f}_i)$, we have:

$$\mathcal{L}_{aux} = -\frac{1}{I} \sum_{i=1}^{I} \log \frac{\exp(\mathbf{q}_i^{a\top} \mathbf{q}_i^{b}/\tau)}{\exp(\mathbf{q}_i^{a\top} \mathbf{q}_i^{b}/\tau) + \sum_{j=1}^{I} \exp(\mathbf{q}_j^\top \mathbf{q}_i/\tau)\, \mathbf{1}(j \neq i)}. \tag{7}$$

Ultimately, we obtain the overall optimization objective of MPDC as:

$$\mathcal{L} = \lambda_1 \mathcal{L}_{clu} + (1 - \lambda_1)\mathcal{L}_{emb} + \lambda_2 \mathcal{L}_{aux} + \lambda_3 \mathbb{E}_i\left(\mathcal{H}\left(q_\theta(n \mid \mathbf{f}_i)\right)\right), \tag{8}$$

where different $\lambda$ are hyperparameters for balancing the loss. Eq. 8 indicates that the pattern categories generated by the clustering component guide the construction of the OOD hyperspherical embedding space. Conversely, learning instance-specific semantics in the embedding component enhances the stability and validity of the clustering space. The training process is shown in Alg. 1.

### 4.3 DETECTION AND ACCOMMODATION

After training the MPDC module on the motion pattern set $M$, the first step involves determining a set of OOD detection thresholds $\Gamma = \{\gamma_0, ..., \gamma_N\}$. Specifically, MPDC feeds the samples $\mathbf{x}_i$ into the clustering component to obtain the category assignment. Based on this, samples are routed to the appropriate embedding heads. Using the vMF distribution in Eq. 5, the inner product between the embedding $f_\theta^n(\mathbf{f}_i)$ and the prototype $\mathbf{z}_n$ in space $n$ is computed as a distance metric, denoted as:

$$\mathcal{D}_n = \left\{ i \mid \arg\max_n q_\theta(n \mid \mathbf{f}_i) = n \right\}, \quad d_i = f_\theta^n(\mathbf{f}_i)^\top \mathbf{z}_n \quad \text{for } i \in \mathcal{D}_n. \tag{9}$$

For each pattern $n$, GMPDR determines the OOD detection threshold $\gamma_n$ by ranking all embedding distances in descending order and taking the *p-quantile* value. Through these thresholds, GMPDR

further employs a dual-criteria OOD detection mechanism, where each test sample is augmented multiple times. The first criterion requires that augmented versions are clustered into the same motion pattern. The second criterion requires that the distance to the corresponding prototype is greater than the OOD threshold. A test sample is identified as an ID sample by GMPDR only if both criteria are satisfied. Meanwhile, the MPDC module selects a set of representative samples from $M$ for replay. Following Eq. 9, GMPDR selects either random samples or nearest neighbors of the prototype for each motion pattern. This memory selection strategy effectively preserves the semantic diversity of the original dataset. In GMPDR, a replay ratio of 1% is sufficient to capture these highly representative samples, helping mitigate forgetting while maintaining sparsity.

During inference, GMPDR leverages the MPDC modules to determine whether a test sample is novel. If the sample is rejected by all existing patterns, we accumulate the detection results. GM-PDR triggers a switching mechanism when either a sufficient number of OOD instances have been accumulated or when the proportion of OOD within a batch is significant. Subsequently, GMPDR transitions into accommodation, integrating replay samples with the novel dataset to optimize the model and generating a new MPDC module. Once the new threshold is configured and new replay samples are selected, the model reverts to inference. This process is detailed in Alg. 2.

## 5 EXPERIMENT

### 5.1 EXPERIMENT SETTING

**Datasets.** We build the OWPTP paradigm on three complex and diverse datasets: SDD (Robicquet et al., 2016), ETH (Pellegrini et al., 2009), and UCY (Leal-Taixé et al., 2014). To design tasks for OWPTP, it is necessary to divide and reorganize the datasets. Previous research has indicated that pedestrian motion patterns are related to their respective scenes (Yang et al., 2022; Wu et al., 2022). Therefore, following the dataset partition proposed by Yang et al. (2022), we construct an SDD task comprising four pattern sets and an ETH/UCY task comprising three pattern sets.

**Baselines.** We select state-of-the-art pedestrian trajectory prediction methods as baselines, with which GMPDR is integrated and compared, including ExpertNet (Zhao & Wildes, 2021), PEC-Net (Mangalam et al., 2020), MemoNet (Xu et al., 2022), YNet (Mangalam et al., 2021), PPT (Lin et al., 2024a), and NSP (Yue et al., 2022). We also adopt continual trajectory prediction methods, CL-ER and CL-SGR (Wu et al., 2022). SHELS (Gummadi et al., 2022) is among the few methods that integrate accommodation and detection capabilities simultaneously, although it is initially developed for image classification tasks. We adapt and optimize SHELS to fit the OWPTP paradigm.

**Evaluation metrics.** Following prior works, we use the past 8 frames to predict the future 12 steps, and generate 20 future trajectories for each instance, selecting the one that best matches the ground truth (GT). Average Displacement Error (ADE) and Final Displacement Error (FDE) measure the average position distance and the endpoint distance between predictions and GT. For accommodation, we introduce Final-ADE/FDE (FADE and FFDE), which evaluate across all data after learning all patterns, reflecting the final performance. Incremental-ADE/FDE (IADE and IFDE) compute the average performance after each learning phase, reflecting the performance during learning. We also calculate the forgetting degree (FGT). For detection, we present the OOD detection AUROC and Novelty Detection Rate when new motion patterns are introduced.

**Implementation details.** We predefine 5 bases per motion pattern set, employ LoRAs of rank 4, and set the embedding dimension to 128. The temperature $\tau$ is 0.5, and the balanced hyperparameters $\lambda$ of 0.5 are applied across all components. We set the OOD threshold set $\Gamma$ at the 70th percentile of distances; the replay ratio is maintained at 1%. More details are provided in Appendix F.

### 5.2 PERFORMANCE OF ACCOMMODATION AND DETECTION

Table 1 presents the accommodation performance of GMPDR and baselines on the SDD task under the OWPTP paradigm. Since the baselines lack detection capabilities, we provide the task boundaries manually. Compared with the suboptimal method, GMPDR achieves improvements of 1.24 and 0.89 in FFDE and IFDE, effectively alleviating forgetting. As a goal-based framework, GMPDR consistently enhances performance in FADE and IADE when integrated with different baselines as trajectory refinement modules, outperforming the suboptimal method by 0.48 and

Table 1: Experiments conducted on SDD evaluate accommodation performance under OWPTP. GMPDR is built upon various methods and is validated through averaging across four pattern orders.

| Method | FADE ↓ | FFDE ↓ | IADE ↓ | IFDE ↓ | FGT-A ↑ | FGT-F ↑ |
|---|---|---|---|---|---|---|
| ExpertNet | 14.23 | 14.34 | 13.81 | 14.85 | -3.06 | -1.28 |
| **+GMPDR** | **13.36±0.35** | **11.79±0.43** | **13.32±1.01** | **11.78±0.52** | **-2.02±0.98** | **-0.07±0.62** |
| PECNet | 9.92 | 12.42 | 10.68 | 12.41 | -0.30 | -1.28 |
| **+GMPDR** | **9.74±0.84** | **11.66±0.46** | **10.48±0.31** | **11.82±0.53** | **0.46±0.52** | **-0.08±0.65** |
| MemoNet | 9.91 | 15.37 | 9.49 | 14.33 | -1.49 | -3.61 |
| **+GMPDR** | **8.56±0.25** | **11.70±0.43** | **8.54±0.25** | **11.90±0.52** | **-0.39±0.35** | **-0.07±0.62** |
| YNet | 8.05 | 12.41 | 7.91 | 12.22 | -0.74 | -1.34 |
| **+GMPDR** | **7.38±0.14** | **11.38±0.42** | **7.55±0.40** | **11.51±0.60** | **0.04±0.33** | **-0.12±0.55** |
| PPT | 7.35 | 12.40 | 7.36 | 12.39 | -0.52 | -1.28 |
| **+GMPDR** | **7.04±0.17** | **11.65±0.45** | **7.17±0.27** | **11.81±0.52** | **-0.05±0.35** | **-0.10±0.59** |
| NSP | 6.81 | 12.39 | 6.75 | 12.22 | -0.49 | -1.32 |
| **+GMPDR** | **6.33±0.26** | **11.15±0.51** | **6.39±0.24** | **11.33±0.71** | **-0.09±0.23** | **-0.19±0.52** |

Table 2: Experiments on ETH/UCY, where the unit is meters. Each experiment is performed and averaged in three motion pattern orders, and GMPDR employs YNet for trajectory refinement.

| Method | FADE ↓ | FFDE ↓ | IADE ↓ | IFDE ↓ | FGT-A ↑ | FGT-F ↑ |
|---|---|---|---|---|---|---|
| MemoNet | 0.232 | 0.409 | 0.222 | 0.393 | -0.047 | -0.085 |
| YNet | 0.246 | 0.417 | 0.207 | 0.339 | -0.136 | -0.274 |
| NSP | 0.199 | 0.415 | 0.195 | 0.336 | -0.072 | -0.258 |
| PPT | 0.197 | 0.378 | 0.188 | 0.325 | -0.034 | -0.113 |
| **GMPDR** | **0.186±0.01** | **0.293±0.02** | **0.182±0.02** | **0.280±0.03** | **-0.021±0.01** | **-0.049±0.02** |

Table 3: Compared to CL methods and SHELS on SDD, SHELS and GMPDR are built upon YNet.

| Method | FADE ↓ | FFDE ↓ | IADE ↓ | IFDE ↓ | AUROC ↑ | Novelty Detection Rate ↑ |
|---|---|---|---|---|---|---|
| CL-ER | 13.32 | 15.21 | 13.05 | 15.10 | - | - |
| CL-SGR | 12.98 | 14.85 | 12.78 | 14.75 | - | - |
| SHELS | 7.93 | 12.04 | 7.81 | 11.97 | 0.6201 | 58.3% |
| **GMPDR** | **7.38** | **11.38** | **7.55** | **11.51** | **0.7456** | **87.5%** |

0.36. Table 2 presents the performance comparison between four optimal baselines and GMPDR on the ETH/UCY task, where GMPDR also demonstrates superior performance. These improvements highlight the effectiveness of the representative sparse replay mechanism and the compatibility of GMPDR, further validating the importance of focusing on the goal prediction stage in OWPTP.

Existing continual pedestrian trajectory prediction methods lack the ability to autonomously detect novel motion patterns. CL-ER and CL-SGR in Table 3 utilize replay and pseudo-replay mechanisms to achieve continual accommodation. However, because they fail to account for goal prediction, these methods require large replay buffers and exhibit more limited performance compared with GMPDR. Previous open-world learning approaches that aim to achieve accommodation and detection are often rudimentary and primarily limited to supervised image classification. For comparison, SHELS is adapted to the OWPTP scenario and adopts the basic framework of our method. Nevertheless, its accommodation performance is inferior to that of GMPDR, which may be attributed to the challenges its regularization mechanism faces in handling complex trajectory prediction tasks.

For OWPTP, the objective of detection is to recognize the emergence of novelty. As shown in Table 3, GMPDR outperforms SHELS by 0.1245 in terms of AUROC. A set-level metric, Novelty Detection Rate, reflects the model's ability to detect and trigger a switch upon encountering new patterns. GMPDR achieves a success rate of nearly 90%. These results demonstrate that GMPDR possesses more reliable novel motion pattern detection capabilities. In Fig. 4, GMPDR achieves continual detection and accommodation in two tasks. When novel pattern sets are introduced (black boundary), the number of unknown motion trajectories gradually increases in each test batch. Consequently, the proportion of OOD samples detected within a batch rises, eventually triggering the switching condition (red boundary). At this point, GMPDR transitions into the accommodation phase, during which a novel dataset is used to assimilate the novel motion patterns into the ID. The trigger condition employed here is a significant increase in the OOD proportion (a 10% rise). This does not imply that GMPDR is limited to batch-wise inference and detection. Alternative trigger conditions, such as accumulating a certain number of OOD samples and using them as a training set for novel patterns, are also viable. For more experiments, including visual comparisons of prediction results and the adaptation of CL methods to trajectory prediction tasks, refer to the Appendix G.

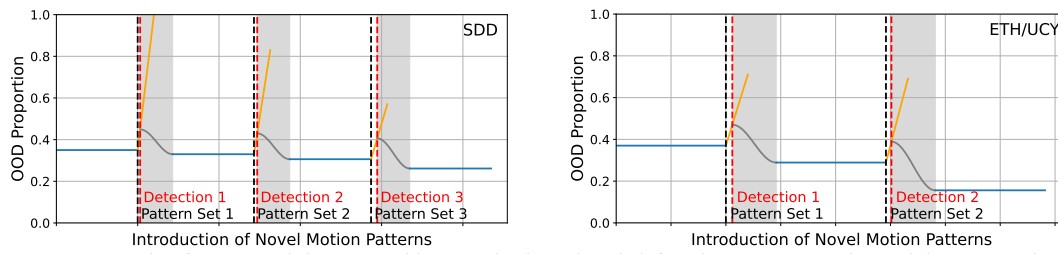

Figure 4: Experiments conducted on two datasets evaluated the ability of GMPDR to maintain continual detection. The proportion of detected OOD instances increases (orange) when novel motion patterns emerge, thereby triggering a transition to the accommodation phase (gray).

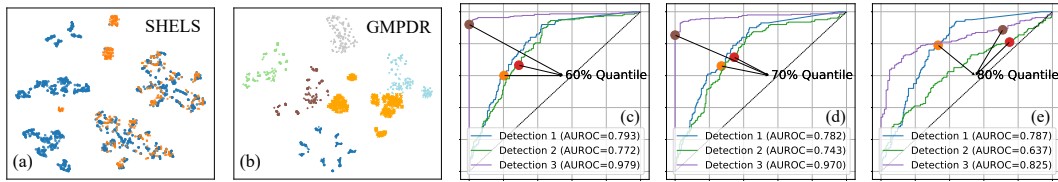

Figure 5: (a)(b): Visualization of the embeddings, with orange representing OOD samples and other colors representing ID samples. (c)(d)(e): ROC curves and p-quantile positions for detection.

Table 4: Impact of replay strategy/ratio, number of bases/motion patterns per set, and LoRA rank.

| Replay | FADE | FFDE | Bases | FADE | FFDE | Rank | FADE | FFDE |
|---|---|---|---|---|---|---|---|---|
| Random 1% | 8.08 | 12.28 | 1 | 7.83 | 12.01 | 2 | 7.86 | 11.64 |
| **GMPDR 1%** | 7.47 | 11.52 | 3 | 7.62 | 11.51 | **4** | 7.40 | 11.39 |
| GMPDR 5% | 7.48 | 11.65 | **5** | 7.49 | 11.42 | 8 | 7.65 | 11.52 |
| GMPDR 10% | 7.38 | 11.49 | 10 | 7.41 | 11.40 | 16 | 7.95 | 11.87 |

## 5.3 Algorithm Analysis

Fig. 5(a)-(b) illustrates the embedding space of GMPDR and SHELS during detection. We observe that GMPDR effectively clusters ID instances into motion patterns while differentiating OOD samples. As part of the dual OOD detection criteria, the p-quantile serves to determine OOD threshold $\Gamma$. We apply 8 augmentations to each sample and require that more than half of the versions be classified into the same cluster. Fig. 5(c)-(e) presents the ROC curves and the corresponding positions of $\Gamma$ selected based on different p-quantiles. The three curves represent the detection of three novelty introductions. Regardless of the p-quantile selected, the AUROC remains high, demonstrating the effectiveness of MPDC for OOD detection. Different replay strategies and ratios are evaluated in the left panel of Table 4. The results indicate that representative replay significantly outperforms random replay, and a sparse ratio of only 1% is sufficient for effective performance. The middle panel of Table 4 illustrates that when the number of bases is limited, MPDC struggles to distinguish among diverse patterns. Consequently, using a larger number of bases achieves more favorable clustering outcomes. However, an excessively large number of bases results in overly sparse sample allocation per cluster. Table 4 also presents the impact of the LoRA rank. Although a higher rank can offer a larger parameter space to learn pattern clustering, an excessively large rank diminishes the effect of regularization, compromising generalization (Lin et al., 2024b). Benefiting from model efficiency, GMPDR maintains a controlled memory footprint. Whenever GMPDR learns a novel pattern set in the ETH/UCY task, the model stores replay samples and the expanded module, resulting in an average memory usage of 14.81 MB in float32. Appendix H provides further algorithm analysis.

## 6 Conclusion

This paper offers insights into endowing trajectory prediction models with autonomous lifelong learning capabilities in open-world environments. Proposed OWPTP paradigm focuses on detecting and accommodating novel motion patterns. We analyze the essence of novel motion and emphasize the critical role of the goal prediction stage. Building on this foundation, the proposed GMPDR is a highly compatible framework that employs the MPDC module to generate abstract clustering and hyperspherical embeddings, enabling OOD detection and representative sparse replay. Future research can explore autonomous construction of novel datasets to further reduce human intervention.

## REPRODUCIBILITY STATEMENT

We provide a complete set of materials to reproduce experiments. Code is anonymously available at: `https://anonymous.4open.science/r/GMPDR-03C3` (a public release is planned after the review period). For transparency, the hyperparameter details for every experiment are reported in Appendix F. Appendix F also describes the task-split and data-processing details.

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

# APPENDIX

## A GOAL-BASED FRAMEWORK ANALYSIS

### A.1 EXPERIMENTAL VALIDATION

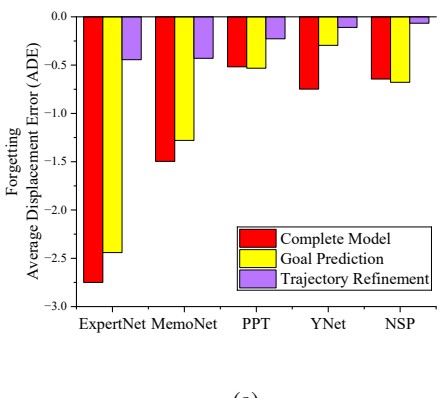

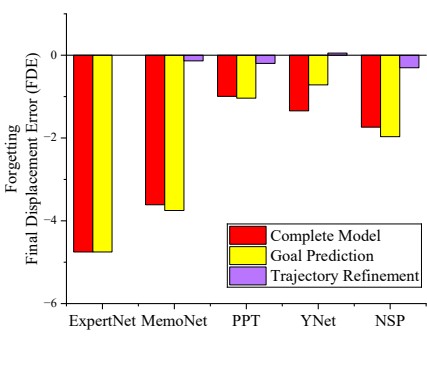

(a)                                                     (b)

Figure 6: Forgetting that occurs after continual accommodation in different pedestrian trajectory prediction methods. (a) Average Displacement Error forgetting. (b) Final Displacement Error forgetting.

| Motion Pattern Set | Goal Prediction $\mathcal{M}_{goal}$ | Trajectory Refinement $\mathcal{M}_{traj}$ |
|---|---|---|
| Pattern set 1 | 0.2626 | 0.0664 |
| Pattern set 2 | 0.2713 | 0.0107 |
| Pattern set 3 | 0.2302 | 0.0528 |

Table 5: Relative parameter changes that occur in NSP during the acquisition of novel motion patterns. Each row reflects the parameter changes following accommodation. It is evident that a greater degree of changes occurs in the goal prediction module.

To enhance OOD detection performance and effectively localize catastrophic forgetting, we focus on the discriminative information inherent in motion patterns by emphasizing the goal prediction module. Our empirical findings confirm the critical role of the goal prediction module in this endeavor. To further validate its importance, we decouple a wide range of pedestrian trajectory prediction methods into two distinct modules: goal prediction and trajectory refinement. We then apply these methods to continually learn various motion patterns and measure the resulting performance degradation, represented by the red portion in Fig. 6. This degradation indicates that existing trajectory prediction methods are prone to catastrophic forgetting when learning new motion patterns. When we freeze the trajectory refinement module and optimize only the goal prediction module, the model still exhibits significant forgetting, as shown by the yellow portion in Fig. 6. In contrast, when the goal prediction module is frozen and only the trajectory refinement module is optimized, the extent of forgetting across different methods remains relatively controlled, as indicated by the purple portion in Fig. 6.

We compare Fig. 6 (a) and (b) side by side. Fig. 6 (a) illustrates the average displacement error forgetting, reflecting the overall accuracy of the predicted trajectory. Consequently, the trajectory refinement module shows a slight decline in performance over time. In contrast, Fig. 6 (b) presents the final displacement error forgetting, which primarily reflects the accuracy of the pedestrian's destination prediction and is therefore largely determined by the goal prediction stage.

The experiments in Fig. 6 validate the importance of the goal prediction component from a performance perspective. We further validated this from the perspective of parameter changes. Table 5

illustrates the extent of relative parameter changes each time the NSP (Yue et al., 2022) method acquires a novel set of motion patterns. Larger relative changes indicate more significant shift of parameter space and a greater susceptibility to catastrophic forgetting. Notably, the magnitude of parameter change is larger during the goal prediction stage. As a state-of-the-art hybrid model, NSP employs neural networks to approximate PDEs for trajectory refinement. Specifically, NSP simulates the environmental and social forces exerted by both the scene and the crowd during movement, which subsequently influence pedestrian acceleration. This mechanism reflects the adherence of trajectory refinement module to explicit physical principles, thereby facilitating more effective knowledge transfer across different motion patterns.

Based on the previous analysis and the aforementioned experimental results, the goal-based framework facilitates the identification and mitigation of forgetting. Furthermore, it indicates that the discriminative nature of novel motion patterns mainly arises from epistemic uncertainty. We emphasize the importance of the goal prediction stage, but this does not mean that the trajectory refinement module should be completely ignored. However, we highlight that the goal prediction stage and the corresponding goal-based framework are the key bottlenecks of the OWPTP problem.

## A.2 FRAMEWORK ADVANTAGES

In the detection phase, the goal-based framework highlights the strong correlation between motion patterns and trajectory goals. By modeling the relationship between trajectory instances and intended goals, this framework enables the extraction of trajectory features that are rich in motion pattern information. With these informative features, the implementation of motion pattern OOD detection becomes more feasible.

In the accommodation phase, the goal-based framework reduces the reliance on large volumes of replay samples. Previous continual pedestrian trajectory prediction approaches depended heavily on storing extensive replay samples (Yang et al., 2022; Wu et al., 2022), primarily to capture social interaction information. Specifically, these methods required multiple sets of pedestrian trajectories simultaneously to simulate interactions and avoidance behaviors among agents–functions typically handled by the trajectory refinement module. In contrast, the goal-based framework focuses on the goal prediction module and transfers the knowledge required by the trajectory refinement module across different patterns. Consequently, it does not require access to a large number of simultaneous pedestrian trajectories for knowledge replay. This characteristic allows the goal-based framework to inherently avoid dependence on large memory buffers, thereby supporting sparse and representative replay strategies.

## B  GOAL PREDICTOR ARCHITECTURE

To capture motion features and learn epistemic uncertainty, GMPDR first implements the fundamental mission of goal prediction. Inspired by prior work (Mangalam et al., 2021; Yue et al., 2022), GMPDR employs three sub-networks based on the classical U-Net architecture (Ronneberger et al., 2015): the input encoder $U_e$, the goal decoder $U_g$, and the auxiliary trajectory decoder $U_t$.

As the focus lies on goal prediction rather than trajectory refinement, GMPDR only requires the past observation trajectory $\mathbf{R}_{obs}^{(i)} = \left[\mathbf{r}_0, \ldots, \mathbf{r}_k\right]$ and the complementary information $S_{goal}^{(i)}$ as input $\mathbf{x}_i$ to predict the goal, i.e., $\mathbf{x}_i = \{\mathbf{R}_{obs}^{(i)}, S_{goal}^{(i)}\}$. Here, the complementary information $S_{goal}^{(i)}$ refers to a pixel-level semantic segmentation map of the scene $\mathcal{I}^{(i)}$ with dimensions $H \times W$. To align multimodal inputs effectively, one intuitive approach is to encode the scene segmentation map using a neural network. However, prior research has shown that encoding scene information can distort the original spatial structure, thereby complicating its integration with trajectory coordinates (Mangalam et al., 2021; 2020). To address this issue, we represent the observed trajectory as heatmaps of the same size as the scene, where each value is inversely proportional to the distance from the pixel to the corresponding trajectory point. As a result, the heatmap will exhibit a distinct peak at the coordinates corresponding to the original trajectory point, with a gradual decline in all directions. Formally, for a timestamp $t$ satisfying $0 \leq t \leq w_{obs}$, the corresponding heatmap $\mathbf{H}_t$ is:

$$\mathbf{H}_t(i,j) = 2 \cdot \frac{\|(i,j) - \mathbf{r}_t\|}{\max\limits_{(x,y) \in \mathcal{I}} \|(x,y) - \mathbf{r}_t\|}. \tag{10}$$

By stacking the trajectory heatmaps along the temporal dimension and combining them with the scene segmentation map, we unify the input while preserving the original spatial structure. These processed inputs are then fed into the encoder $U_e$ for feature encoding. Each block of $U_e$ reduces the spatial dimension by half through max pooling, followed by convolutional operations and non-linear ReLU activation to increase the channel depth. Through this hierarchical processing, $U_e$ extracts both deep and multi-scale features, which are then passed to the decoder.

The goal decoder $U_g$ mirrors the expansion arm of the U-Net decoding path. Specifically, $U_g$ performs upsampling via transposed convolution at each level, concatenates the resulting feature maps with the corresponding encoder outputs, and subsequently fuses them using convolutional layers. The final output of $U_g$ is a pixel-level goal probability distribution map $P_{goal}(\mathbf{x}_i)$ of size $H \times W$, obtained after applying a sigmoid activation function. Although the coordinate with the highest probability value could be directly selected as the predicted goal, the probability distribution generated during the early stages of training may be unstable. To enhance the robustness of the sampling, the *softargmax* operation is employed to approximate the goal in a probabilistically differentiable manner (Mangalam et al., 2021):

$$\text{softargmax}(P_{goal}) = \left( \sum_i i \cdot \frac{\sum_j e^{P_{goal}(ij)}}{\sum_{i,j} e^{P_{goal}(ij)}}, \ \sum_j j \cdot \frac{\sum_i e^{P_{goal}(ij)}}{\sum_{i,j} e^{P_{goal}(ij)}} \right). \tag{11}$$

Since both $U_e$ and $U_g$ are randomly initialised during the initial learning phase of OWPTP, $U_t$ is introduced as an auxiliary trajectory decoder to stabilise and facilitate the learning process for goal prediction. $U_t$ adopts the same network structure as $U_g$, with the key distinction that it utilises both the encoder features and the goal probability distribution map in skip connection. As a result, $U_t$ predicts the trajectory probability distribution map $P_t(\mathbf{x}_i)$, where $w_{obs} + 1 \leq t \leq w_{obs} + w_{fut}$. By incorporating $U_t$ into the training process, auxiliary trajectory information influences both the encoder and the goal decoder through gradient backpropagation. This mechanism enables the model to better capture motion patterns and enhances the discriminative capacity of deep features.

Given that all outputs are explicit probability distributions, a fixed 2D Gaussian kernel is applied to transform real future trajectories into heatmaps, denoted as $\hat{P}_t(\mathbf{x}_i)$. Subsequently, using the binary cross-entropy (BCE) and loss balance hyperparameter $\lambda$, we formulate the following optimization objective with sample number $I$:

$$\mathcal{L} = -\frac{1}{I} \sum_{i=1}^{I} (\text{BCE}(P_{goal}(\mathbf{x}_i), \hat{P}_{w_{obs}+w_{fut}}(\mathbf{x}_i)) + \lambda \sum_{t=w_{obs}+1}^{w_{obs}+w_{fut}-1} \text{BCE}(P_t(\mathbf{x}_i), \hat{P}_t(\mathbf{x}_i))). \quad (12)$$

By optimizing Eq. 12, the basic GMPDR framework gains the ability to predict goal distributions, which precisely characterises epistemic uncertainty.

In the subsequent accommodation phase, GMPDR freezes the parameters of encoder $U_e$, and focuses solely on optimizing $U_g$ to adapt to the prediction of the newly introduced motion pattern. During this phase, GMPDR performs joint training using both replay samples and the novel motion pattern data, applying Eq. 12 to update $U_g$ while effectively mitigating catastrophic forgetting. The discriminative feature learning of the novel pattern is carried out by the corresponding MPDC module in conjunction with the newly introduced LoRA, as detailed in Subsection 4.2. As previously stated, the auxiliary trajectory decoder $U_t$ can be further optimized, and it continues to play a supportive role in the overall optimization process.

## C LEMMA

We introduce Lemma 1 to ensure that the underlying assumption of set-level contrastive learning are satisfied. If different bases yield identical probabilities, it becomes necessary to eliminate the possibility that two bases represent the same category. Therefore, there should be differences between each cluster or base.

**Lemma 1** (Asymptotic Form of Contrastive Loss (Wang & Isola, 2020)). *Consider a measurable encoder function $f : \mathbb{R}^n \to \mathbb{S}^{m-1}$, a given data distribution $p_{\text{data}}(x)$ and a positive-pair distribution $p_{\text{pos}}(x, y)$ satisfying:*

*1. Symmetry: For all $x, y$,*
$$p_{\text{pos}}(x, y) = p_{\text{pos}}(y, x).$$

*2. Matching marginal: For all $x$,*
$$\int p_{\text{pos}}(x, y) dy = p_{\text{data}}(x).$$

*Given a temperature parameter $\tau > 0$, define the contrastive loss as:*

$$L_{\text{contrastive}}(f; \tau, A) = \mathbb{E}_{(x,y) \sim p_{\text{pos}}, \{x_i^-\}_{i=1}^A \overset{\text{i.i.d.}}{\sim} p_{\text{data}}} \left[ -\log \frac{e^{f(x)^\top f(y)/\tau}}{e^{f(x)^\top f(y)/\tau} + \sum_{i=1}^M e^{f(x_i^-)^\top f(y)/\tau}} \right].$$

*As the number of negative samples $A \to \infty$, the following asymptotic form holds:*

$$\lim_{A \to \infty} \left[ L_{\text{contrastive}}(f; \tau, A) - \log M \right] = -\frac{1}{\tau} \mathbb{E}_{(x,y) \sim p_{\text{pos}}} \left[ f(x)^\top f(y) \right]$$
$$+ \mathbb{E}_{x \sim p_{\text{data}}} \left[ \log \mathbb{E}_{x^- \sim p_{\text{data}}} \left[ e^{f(x^-)^\top f(x)/\tau} \right] \right] \quad (13)$$

*We have the following results: The first term is minimized when $f$ is well aligned with the positive samples, and the second term converges if there exists a uniform encoders.*

## D PROOF OF EQ. 4

Based on the prior work (Shen et al., 2021), we can derive the ELBO of Eq. 4 with Jensen's inequality as follows:

$$\log p_\theta(i \mid \mathbf{f}_i) = \log \sum_{n=1}^N p_\theta(i, n \mid \mathbf{f}_i)$$
$$= \log \sum_{n=1}^N p_\theta(i \mid \mathbf{f}_i, n) \, p_\theta(n \mid \mathbf{f}_i) \frac{q_\theta(n \mid \mathbf{f}_i)}{q_\theta(n \mid \mathbf{f}_i)}$$
$$= \log \mathbb{E}_{q_\theta(n \mid \mathbf{f}_i)} \left[ p_\theta(i \mid \mathbf{f}_i, n) \frac{p_\theta(n \mid \mathbf{f}_i)}{q_\theta(n \mid \mathbf{f}_i)} \right]$$
$$\geq \mathbb{E}_{q_\theta(n \mid \mathbf{f}_i)} \left[ \log p_\theta(i \mid \mathbf{f}_i, n) \right] + \mathbb{E}_{q_\theta(n \mid \mathbf{f}_i)} \left[ \log \frac{p_\theta(n \mid \mathbf{f}_i)}{q_\theta(n \mid \mathbf{f}_i)} \right]$$
$$= \mathbb{E}_{q_\theta(n \mid \mathbf{f}_i)} \left[ \log p_\theta(i \mid \mathbf{f}_i, n) \right] - \text{KL} \left( q_\theta(n \mid \mathbf{f}_i) \,\|\, p_\theta(n \mid \mathbf{f}_i) \right), \quad (14)$$

where KL denotes the Kullback-Leibler (KL) divergence.

# E ALGORITHM PSEUDOCODE

---

**Algorithm 1** MPDC Training Algorithm for Task $T$ with Motion Pattern Set $M$

---

**Input**: Datasets $\mathcal{D}^{(T)}$; Frozen input encoder $U_e$; Predefined motion pattern number $N$; Temperature $\tau$; Epoch for MPDC $epoch$, etc.

1: *# Training MPDC*
2: Initialize task-specific LoRA weights; clustering projector $f_\theta$ and clustering bases $\Omega$; $N$ embedding heads $f_\theta^n$ and prototypes $\mathbf{z}_n$; auxiliary mapping head $f_\theta^{aux}$.
3: **repeat**
4:      Randomly select a batch from $\mathcal{D}^{(T)}$
5:      Obtain augmented feature sets $F^a$ and $F^b$ by $U_e$ and task-specific LoRA weights
6:      Perform clustering and compute $q_\theta(n \mid \mathbf{f}_i)$ by Eq. 2
7:      Calculate the clustering loss $\mathcal{L}_{clu}$ by Eq. 3
8:      Construct an embedding space for each head $f_\theta^n$ based on the categorical prior
9:      Calculate the embedding loss $\mathcal{L}_{emb}$ by Eq. 6
10:      Calculate the auxiliary loss $\mathcal{L}_{aux}$ by Eq. 7
11:      Update MPDC module by Eq. 8
12: **until** *convergence or reaching epoch*

**Output**: MPDC module for Task $T$ with Motion Pattern Set $M$

---

---

**Algorithm 2** GMPDR workflow in the OWPTP paradigm

---

**After learning motion pattern set** $T-1$: Goal prediction model $\mathcal{M}^{T-1}$; MPDC modules $\mathcal{U}_{T-1} = \{\sqcap_0, ..., \sqcap_{T-1}\}$; Previous replay sample set $\mathcal{D}_{replay}^{T-1}$; Previous OOD threshold $\{\Gamma_0, ..., \Gamma_{T-1}\}$, etc.

1: *# Inference*
2: Detection and prediction, ID: $0 : T-1$
3: **repeat**
4:      Augment a test sample or batch of test samples
5:      Detecting augmented samples in each MPDC module using Eq. 9 and OOD thresholds
6:      **if** Samples belong to ID **then**
7:          Predict trajectories using Goal prediction model $\mathcal{M}^{T-1}$
8:      **else**
9:          Cumulative OOD detection results
10:      **end if**
11: **until** *Sufficient OOD samples are accumulated to meet the switching criteria*
12: Switch to training phase
13: *# Training*
14: Accommodation for $T$
15: Mix novel motion pattern data $\mathcal{D}^T$ and replay samples $\mathcal{D}_{replay}^{T-1}$
16: Update the prediction model $\mathcal{M}^{T-1} \to \mathcal{M}^T$ through Eq. 12
17: Create a MPDC module $\sqcap_T$ for pattern set $T$ using Algorithm 1, $\mathcal{U}_{T-1} \to \mathcal{U}_T$
18: Calculate the OOD threshold $\Gamma_T = \{\gamma_0, ..., \gamma_N\}$ using Eq. 9 and p-quantile, integrated as $\{\Gamma_0, ..., \Gamma_{T-1}, \Gamma_T\}$
19: Select the replay sample using Eq. 9, integrated as $\mathcal{D}_{replay}^{T-1} \to \mathcal{D}_{replay}^T$
20: Switch back to inference phase

---

# F    EXPERIMENT SETTING

We construct the OWPTP paradigm on three widely used datasets: SDD (Robicquet et al., 2016), ETH (Pellegrini et al., 2009), and UCY (Leal-Taixé et al., 2014). The Stanford Drone Dataset is one of the most popular benchmarks and is characterised by its complex and diverse motion patterns. It is captured by a drone camera from a bird's-eye view and comprises 5,232 pedestrian trajectories across six distinct scenes. The ETH/UCY dataset is a combination of the ETH and UCY datasets and contains five different scenes comprising approximately 1,500 pedestrian trajectories. To design tasks for continual prediction in open-world environments, we incrementally structure the datasets based on motion patterns. Following the dataset partition proposed by Yang et al. (2022), we construct an SDD task comprising four motion pattern sets and an ETH/UCY task comprising three motion pattern sets, excluding scenes with limited data availability. Table 6 and Table 7 show the task division of the two datasets by scene, respectively. In addition, 20% of the training samples are set aside as the validation set.

| Dataset | Training set | Test set |
|---------|-------------|----------|
| coupa | coupa: 3 | coupa: 0,1 |
| gates | gates: 0,1,3,4,5,6,7,8 | gates: 2 |
| hyang | hyang: 4,5,6,7,9 | hyang: 0,1,3,8 |
| nexus | nexus: 0,1,2,3,4,7,8,9 | nexus: 5,6 |

Table 6: SDD task divides into four motion pattern set.

| Dataset | Training set | Test set |
|---------|-------------|----------|
| ETH | biwi_eth_train, biwi_hotel_train | biwi_eth_val, biwi_hotel_val |
| STU | students001_train, students003_train, uni_examples_train | students001_val, students003_val, uni_examples_val |
| ZARA | crowds_zara01_train, crowds_zara02_train, crowds_zara03_train | crowds_zara01_val, crowds_zara02_val, crowds_zara03_val |

Table 7: ETH/UCY task divides into four motion pattern set.

For performance evaluation, we refer to classic prediction metrics: Average Displacement Error (ADE) and Final Displacement Error (FDE), which measure the average positional distance and the endpoint distance between the predicted trajectory and the GT, respectively. For simplicity, let us assume that we are given $I$ pedestrian trajectories and are required to predict $L$ future frames. When the prediction and GT are denoted by $\mathbf{y}_i^t$ and $\hat{\mathbf{y}}_i^t$, the two evaluation metrics can be formulated as follows:

$$\text{ADE} = \frac{1}{IL} \sum_{i=1}^{I} \sum_{j=1}^{L} \left\| \hat{\mathbf{y}}_i^j - \mathbf{y}_i^j \right\|_2 \tag{15}$$

$$\text{FDE} = \frac{1}{I} \sum_{i=1}^{I} \left\| \hat{\mathbf{y}}_i^L - \mathbf{y}_i^L \right\|_2 . \tag{16}$$

Under the OWPTP, we introduce four new metrics: Final-ADE (FADE) and Final-FDE (FFDE), which evaluate ADE and FDE performance across all data after the model has learned all motion patterns, reflecting the final model performance. In contrast, Incremental-ADE (IADE) and Incremental-FDE (IFDE) compute the average performance after each learning phase, reflecting the model's performance during the learning process. Lower values across all four metrics indicate superior prediction performance. Assume there are $T$ sets of motion patterns or tasks. $\text{ADE}_t$ represents the ADE performance of pattern sets from $0$ to $t$ after task $t$ is accommodated . These OWPTP metrics can be formally expressed as:

$$\text{FADE} = \text{ADE}_T \tag{17}$$

$$\text{FFDE} = \text{FDE}_T \tag{18}$$

$$\text{IADE} = \frac{1}{T} \sum_{t=1}^{T} \text{ADE}_t \tag{19}$$

$$\text{IFDE} = \frac{1}{T} \sum_{t=1}^{T} \text{FDE}_t \tag{20}$$

Additionally, we calculate the forgetting degree (FGT) for ADE and FDE. Specifically, we compute the difference between the performance of each pattern set immediately after accommodation and its performance following the completion of learning all tasks. This difference reflects the extent of performance degradation. Since FGT-A and FGT-F are typically negative, smaller values indicate a greater degree of forgetting.

We employ AUROC as the evaluation metric for instance-based OOD detection performance. It is important to note that the OWPTP task requires detecting multiple novel motion patterns, each of which is mutually exclusive with the others. Consequently, the reported AUROC represents the average performance across all these incremental and individual detection tasks. In Appendix G.4, we present a comparative analysis of GMPDR's AUROC results against those of other methods, while Appendix H examines the influence of different OOD detection criteria.

For the goal prediction model in GMPDR, we use the same U-Net architecture and training settings as Ynet (Mangalam et al., 2021), employing task-specific LoRA weights of rank 4. The prediction model is trained using the Adam optimizer with a learning rate of 0.0001. Meanwhile, no fine-tuning is performed on the pre-trained scene segmentation model. Each set of motion patterns requires 50 epochs to accommodate.

Regarding the MPDC module, we predefine 5 categories per motion pattern set, corresponding to the number of clustering bases and embedding heads. The clustering head, embedding head, and auxiliary head are each implemented as single-layer MLP with normalisation layers. The input dimension of MLP is the flattened dimension of the deepest layer features of the encoder $U_e$, and the feature embedding dimension is set to 128. The temperature parameter $\tau$ used in the contrastive learning loss is 0.5, and a balanced hyperparameter $\lambda$ of 0.5 is applied across all loss components in Eq. 8. For the heuristic selection strategy of the OOD threshold, we define the threshold $\gamma$ as MPDC 70th percentile of distances, and the replay ratio is maintained at 1%. The MPDC module is also optimized using the Adam optimizer with a learning rate of 0.0001. For the SDD task, a batch size of 32 and 500 training epochs are employed; for the ETH/UCY task, a batch size of 128 and 50 epochs are used.

For all other methods, we adapt their original code to ensure compatibility with the OWPTP paradigm. The baselines utilise the hyperparameters specified in their respective source code repositories and are appropriately fine-tuned to align with the continual accommodation setting.

## G SUPPLEMENTARY COMPARATIVE EXPERIMENTS

### G.1 VISUALIZATION RESULTS OF GMPDR

We first present the visualization results of GMPDR and the baseline pedestrian trajectory prediction method. Fig. 7 provides a visual comparison of GMPDR and YNet on the ETH/UCY dataset. After the model has learned all patterns, we re-predict the trajectories that were previously learned. It is evident that YNet suffers from forgetting, resulting in prediction failures, as shown by frequent deviations of blue trajectories from the yellow GT. In contrast, GMPDR alleviates forgetting through representative replay, as it continues to produce trajectory predictions that remain highly consistent with the GT.

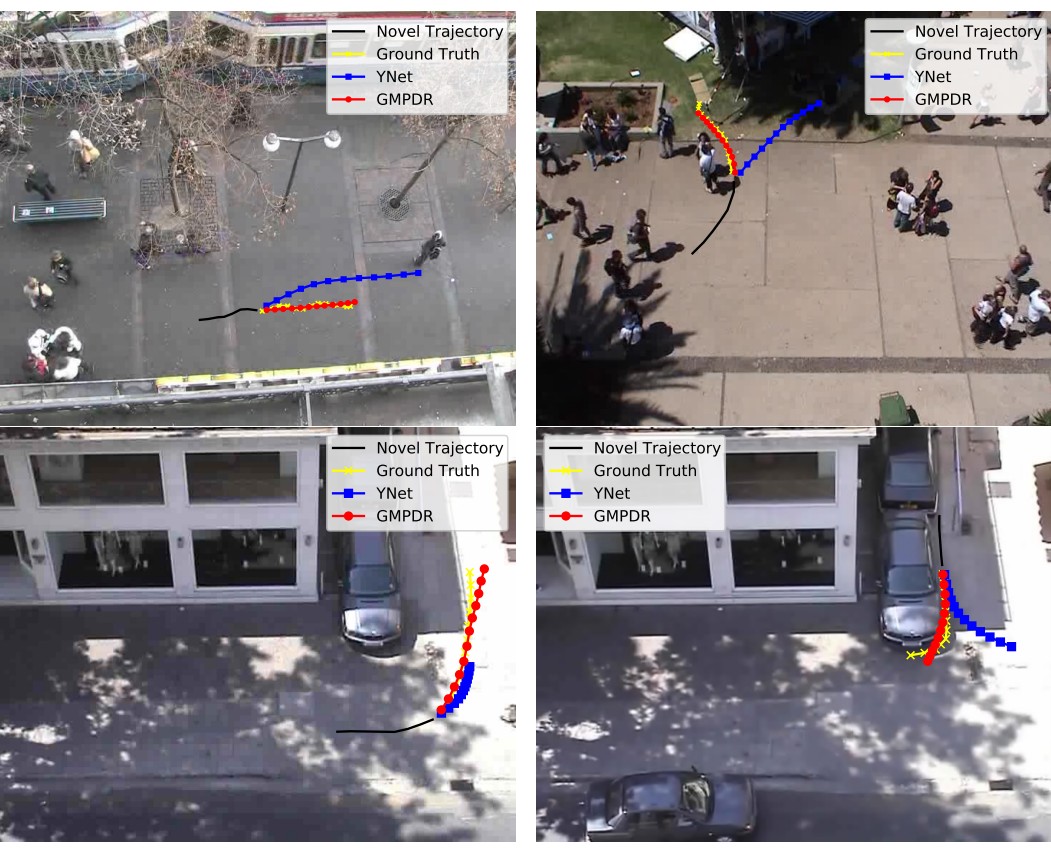

Figure 7: After accommodating a series of motion patterns, GMPDR acquires sufficient knowledge to perform accurate predictions (red trajectories), whereas classical prediction methods tend to suffer from forgetting (blue trajectories).

Furthermore, we also presented more continual detection results graphs of GMPDR. Fig. 8 and 9 demonstrate that GMPDR achieves continual detection and accommodation in the SDD and ETH/UCY tasks, with four and three distinct motion pattern set learning orders illustrated.

These experiments demonstrate that GMPDR successfully detects novelty through the multi-head hyperspherical OOD detection mechanism. This effective detection function, in turn, serves as a switching signal for the accommodation phase. The final introduction and detection shown in Fig. 8(c) is the only detection failure case observed, which may be because the patterns in the final motion pattern set are potentially subsumed by the previous ones. It is important to reiterate that motion patterns are inherently abstract concepts. The division based on scene information in our experiments is merely an approximation of an ideal scenario. When deployed in real-world environments, GMPDR is expected to effectively detect and accommodate novel trajectories.

In addition, an interesting observation is that as GMPDR learns an increasing number of patterns, the proportion of OOD samples per batch in the data stream decreases notably. We attribute this

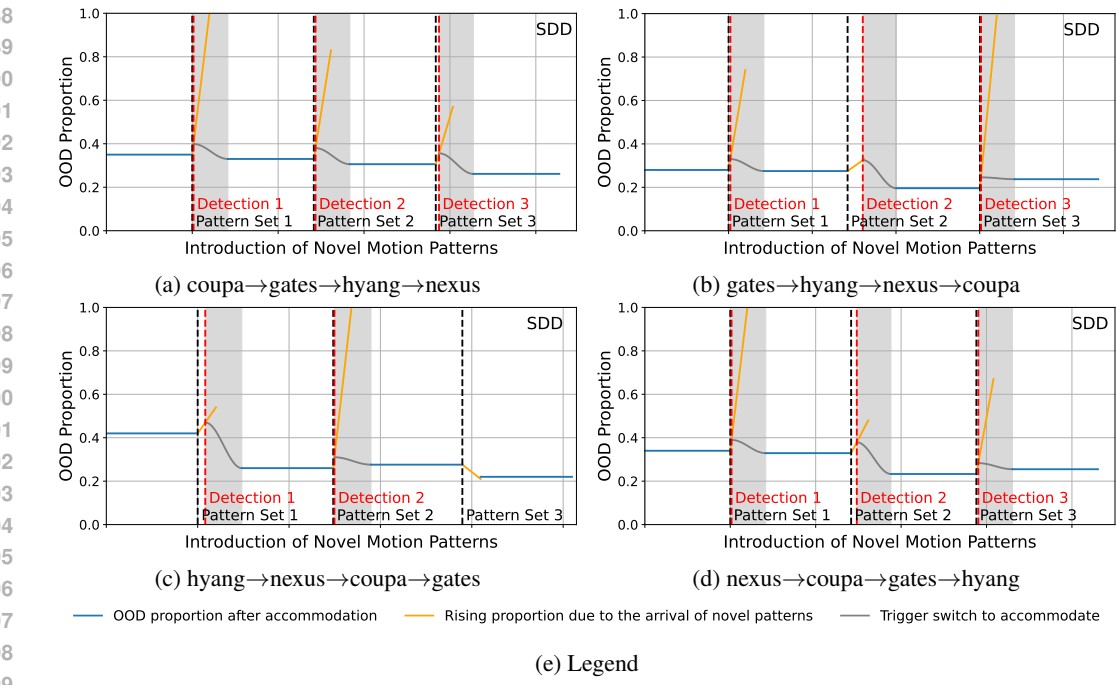

Figure 8: Experiments conducted on SDD evaluated the ability of GMPDR to maintain continual detection under OWPTP.

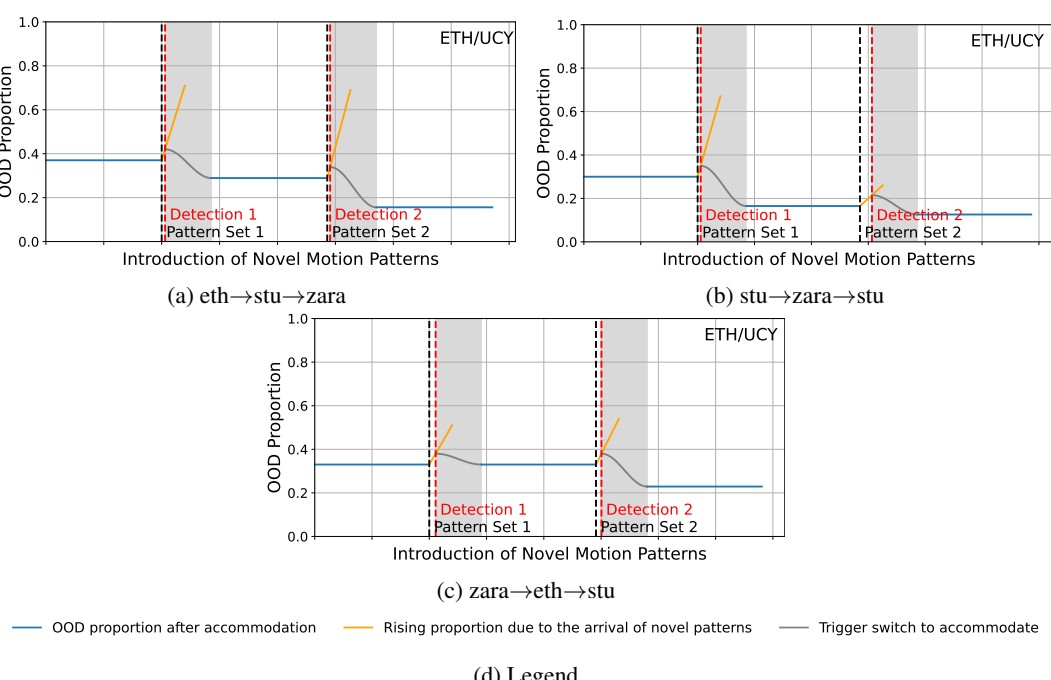

Figure 9: Experiments conducted on ETH/UCY evaluated the ability of GMPDR to maintain continual detection under OWPTP.

trend to the model's gradual accumulation of knowledge through continual accommodation. During the detection phase, each sample is evaluated for its membership across all previously identified patterns. Consequently, newly acquired patterns enhance the earlier pattern set by addressing previously under-learned components, leading to a more stable and well-defined ID over time.

## G.2 COMPARISON WITH CONTINUAL LEARNING METHODS

The accommodation process in OWPTP can be viewed as a CL phase. GMPDR is specifically designed for pedestrian trajectory prediction tasks. To further evaluate the effectiveness of its continual accommodation capability, we adapt several CL methods to pedestrian trajectory prediction and conduct comparative experiments. However, these methods exhibit two key limitations: first, they lack autonomous detection mechanisms, necessitating manual provision of task boundary information during the accommodation process; second, they were originally developed for image classification benchmarks, which may limit their applicability and effectiveness in the context of trajectory prediction.

Table 8: Comparison with the continual learning approach on SDD task, focuses primarily on accommodation performance.

| Method | FADE ↓ | FFDE ↓ | IADE ↓ | IFDE ↓ | FGT-A ↑ | FGT-F ↑ |
|---|---|---|---|---|---|---|
| FineTune | 8.04 | 12.38 | 7.89 | 12.35 | -0.69 | -1.35 |
| EWC | 8.00 | 12.33 | 7.82 | 12.26 | -0.71 | -1.30 |
| iCaRL | 8.22 | 12.91 | 7.95 | 12.46 | -1.42 | -1.34 |
| DER | 7.65 | 12.10 | 7.74 | 12.34 | -0.38 | -0.60 |
| MEMO | 7.73 | 12.01 | 7.70 | 12.25 | -0.54 | -0.97 |
| **GMPDR** | **7.38** | **11.38** | **7.55** | **11.51** | **0.04** | **-0.12** |

We evaluate and adapt several classical and state-of-the-art methods, with results provided in Table 8. All experiments are conducted on the SDD task using YNet as the baseline network to ensure a consistent evaluation framework, under a fixed replay ratio of 1%. FineTune adopt a straightforward fine-tuning strategy without additional constraints. EWC (Kirkpatrick et al., 2017), a representative regularization-based method, mitigates catastrophic forgetting by penalizing changes to important parameters using Fisher information matrices. However, due to the complexity of trajectory prediction networks, accurately estimating parameter importance remains challenging, which limits EWC's effectiveness.

iCaRL (Rebuffi et al., 2017), a classical replay-based approach, selects and preserves highly representative samples for rehearsal. Nevertheless, replay-based methods such as iCaRL encounter difficulties in OWPTP, where motion patterns are inherently unsupervised and cannot be segmented based on category labels as in image classification tasks. This limitation underscores the importance of abstract motion pattern clustering, which is a core focus of GMPDR. In fact, without motion pattern clustering, directly selecting features centered across the entire feature distribution fails to preserve complex semantic information in replay samples, ultimately leading to overfitting of certain trajectories and suboptimal performance. Methods such as DER (Yan et al., 2021) and MEMO (Zhou et al., 2023) represent more advanced replay strategies that incorporate model extensions. However, their effectiveness remains limited. In contrast, GMPDR employs sparse representative replay based on motion pattern clustering, enabling the memory system to retain the semantic meaning of each distinct motion pattern.

We also evaluated classical OWM (Zeng et al., 2019) and the more recent CLDNet (Li et al., 2024). However, these orthogonal space-based update methods require large projection matrices. While such matrices are manageable in image classification tasks, their dimensions become prohibitively large in trajectory prediction scenarios, severely limiting their applicability and transferability.

### G.3 COMPARISON WITH CONTINUAL PEDESTRIAN TRAJECTORY PREDICTION METHODS

Some studies have investigated pedestrian trajectory prediction within a continual learning scenario (Habibi et al., 2020; Knoedler et al., 2022; Yang et al., 2022; Wu et al., 2022). However, as previously noted, these approaches rely on manually defined task boundaries and are unable to automatically identify emerging motion patterns. Consequently, it can be argued that these methods have only addressed the continual accommodation phase of OWPTP.

In the accommodation phase, the limitations of these methods stem from their inability to account for the critical factors of goal prediction and epistemic uncertainty, which results in suboptimal performance. The most recent and advanced continual pedestrian trajectory prediction methods, CLTP-MAN, CL-ER, and CL-SGR, are all based on replay or pseudo-replay mechanisms (Yang et al., 2022; Wu et al., 2022). These approaches typically require large replay buffers, such as 10% of the dataset, to store social interaction information. However, this rapid increase in memory overhead restricts their applicability in open-world environments.

To further evaluate the effectiveness of the proposed GMPDR framework, we specifically examine its performance during the accommodation phase and compared it with these state-of-the-art methods. All performance metrics are based on the results reported in the original papers or obtained from the source code repositories. For the sake of fair comparison, we restrict the baseline methods to using only 1% of the replay samples. Some comparison results have already been presented in the main text.

Since this subfield has not been sufficiently explored, the relevant evaluation frameworks have not yet been fully standardized. Given that CLTP-MAN is not open-source, we consistently adopt its evaluation metrics in this subsection, specifically average error (AER) and average forgetting (FGT). Let $R_{i,j}$ denotes the testing error (ADE/FDE) at the jth task after training on the ith task, $K$ is the total number of tasks. AER is a metric used to assess prediction performance in continual accommodation. FGT evaluates the degree to which a model forgets previously learned knowledge. Lower values indicate better performance for both metrics. Their calculation methods are as follows:

$$\text{AER} = \frac{1}{K(K+1)/2} \sum_{i=1}^{K} \sum_{j \leq i} R_{i,j} \tag{21}$$

$$\text{FGT} = \frac{1}{K(K-1)/2} \sum_{i=2}^{K} \sum_{j < i} R_{i,j} - R_{j,j} \tag{22}$$

Table 9: Comparison with continual pedestrian trajectory prediction methods on ETH/UCY task. Each experiment is performed and averaged in three different motion pattern orders, and GMPDR employs YNet for trajectory refinement.

| Method | AER-ADE ↓ | FGT-ADE ↓ | AER-FDE ↓ | FGT-FDE ↓ |
|---|---|---|---|---|
| CL-ER | 0.545±0.075 | 0.164±0.065 | 1.145±0.165 | 0.345±0.117 |
| CL-SGR | 0.515±0.048 | 0.144±0.034 | 1.105±0.103 | 0.323±0.034 |
| CLTP-MAN | 0.403±0.023 | 0.040±0.010 | 0.823±0.061 | 0.083±0.025 |
| **GMPDR** | **0.185±0.011** | **0.029±0.013** | **0.286±0.018** | **0.058±0.025** |

Table 9 presents the comparison results between GMPDR and continual pedestrian trajectory prediction baselines on the ETH/UCY task. It can be observed that GMPDR achieves superior performance with significantly less forgetting compared to these methods. This suggests that GMPDR effectively identifies the locations where forgetting occurs by focusing on goal prediction and epistemic uncertainty. Furthermore, this mechanism supports a sparse representative replay strategy, which not only mitigates forgetting effectively but also facilitates knowledge transfer. The visualization results in Fig. 10 demonstrate that GMPDR successfully distinguishes trajectory instances through abstract clustering of motion patterns. Selecting samples for replay at this clustering level effectively preserves the semantic structure of the original dataset, thereby validating the effectiveness of the proposed sparse representative replay strategy.

## G.4 COMPARISON WITH OPEN-WORLD LEARNING METHODS

Below, we compare GMPDR with other open-world learning methods. Although several open-world learning approaches have addressed the two phases of continuous detection and accommodation (Zhu et al., 2024; Zeno et al., 2021; Lee et al., 2020), they typically operate as prototype methods applied to image classification benchmark tasks. Direct application of these methods to OWPTP is challenging due to their lack of fundamental trajectory prediction capabilities and inability to recognize distinct motion patterns. To validate the effectiveness of the proposed GMPDR, we conduct a comparative analysis using SHELS (Gummadi et al., 2022) as a case study.

SHELS is a state-of-the-art open-world learning method that effectively captures the interplay between the detection and accommodation phases, corresponding the integration of OOD detection and CL. To adapt SHELS for the OWPTP paradigm, we utilize our proposed goal-based framework and encoder-decoder architecture as the base model. Building upon this structure, we incorporate SHELS's regularization strategy to mitigate catastrophic forgetting during the accommodation phase, and apply its cosine normalization technique to facilitate effective OOD detection. This adapted framework represents a refined and optimized version of the original SHELS method, tailored specifically for OWPTP paradigm.

Table 10: Comparison with the open-world learning approach on SDD task, focuses primarily on accommodation performance.

| Method | FADE ↓ | FFDE ↓ | IADE ↓ | IFDE ↓ | FGT-A ↑ | FGT-F ↑ |
|---|---|---|---|---|---|---|
| YNet | 8.05 | 12.41 | 7.91 | 12.22 | -0.74 | -1.34 |
| SHELS | 7.93 | 12.04 | 7.81 | 11.97 | -0.66 | -1.03 |
| **GMPDR** | **7.38** | **11.38** | **7.55** | **11.51** | **0.04** | **-0.12** |

We compare GMPDR with SHELS on the SDD task. First, we evaluate their performance in knowledge accommodation, as presented in Table 10. It can be observed that SHELS achieves a certain level of improvement over the YNet baseline, primarily due to the regularization strategy alleviating catastrophic forgetting. However, this approach appears insufficient for handling complex trajectory prediction tasks. In contrast, GMPDR employs a sparse representative replay strategy, which more effectively mitigates forgetting. This highlights a key advantage of GMPDR over other open-world learning approaches: under the OWPTP paradigm, sparse representative replay better preserves prior knowledge, which is particularly critical given the inherent complexity of trajectory prediction.

Table 11: Comparison with the open-world learning approach on SDD task, focuses primarily on detection performance.

| Method | Novelty Detection Rate ↑ | ID Pattern Forgetting Rate ↓ | AUROC ↑ |
|---|---|---|---|
| SHELS | 58.3% | 45.8% | 0.6201 |
| **GMPDR** | **87.5%** | **0%** | **0.7456** |

Next, we evaluate the detection performance, as presented in Table 11. To quantitatively assess the model's ability to detect novel motion patterns, we introduce three evaluation metrics. Metrics **Novelty Detection Rate** and **ID Pattern Forgetting Rate** correspond to the set level, considering whether novel motion pattern sets have been detected or known motion pattern sets have been forgotten. Metric **AUROC** corresponds to the instance level, directly focusing on the model's OOD detection performance for distinguishing each instance.

Each time a new motion pattern set is introduced, we examine whether the model successfully identifies the novelty and transitions into the accommodation phase. This transition success rate is referred to as the Novelty Detection Rate. Our results show that GMPDR achieves an 87.5% success rate in detecting novel patterns. This high performance can be attributed to the MPDC module's clustering and embedding capabilities, which enable the model to distinguish between different motion patterns by clustering trajectory samples accordingly. Additionally, the explicit vMF modeling and distance metric ensure that the embeddings of novel trajectories significantly differ from those of ID trajectories. In contrast, SHELS demonstrates a lower detection success rate. This is primarily due to SHELS' limited trajectory aggregation capability, which impedes the formation of trajectory-to-pattern associations. Moreover, its cosine-normalized embedding strategy, used as a

distance metric, lacks prior support and may not be optimal. These findings highlight the superior detection performance of the proposed GMPDR framework compared to other open-world learning approaches.

Another metric that quantifies detection capability is the ID Pattern Forgetting Rate. This refers to the phenomenon wherein previously learned ID patterns are gradually forgotten during subsequent accommodation processes, causing them to be misclassified as OOD patterns. In our specific calculation, we assume that a set of patterns originally identified as ID becomes progressively classified as OOD, with a degradation proportion increasing by 50%. Our observations indicate that SHELS exhibits a significant forgetting rate, requiring the model to relearn previously acquired patterns and thereby incurring additional computational overhead. This issue primarily stems from parameter overwriting within the encoder. In contrast, GMPDR achieves a forgetting rate of 0 for ID patterns due to our strategy of freezing the encoder, which ensures full preservation of the prior feature space. These results further validate the effectiveness of GMPDR for the OWPTP paradigm.

The final metric is the average AUROC value across multiple detections. We first note that GMPDR achieves significantly higher AUROC values compared to SHELS. This further confirms the effectiveness of hyperspherical OOD detection combined with abstract clustering specifically designed for trajectory tasks. Attributing to the intrinsic complexity of trajectory prediction tasks and the abstract characteristics of motion patterns, it is inherently challenging to precisely determine whether each given trajectory instance belongs to the known class. Nevertheless, we emphasize that this does not suggest that effective detection function in OWPTP is unattainable. In OWPTP, the switching mechanism typically depends on cumulative evidence, which allows for a more lenient evaluation of individual samples. Therefore, Novelty Detection Rate and ID Pattern Forgetting Rate are practically more direct metrics for OWPTP.

Transferring prototype open-world learning algorithms to the OWPTP paradigm is challenging. We emphasize that SHELS exhibits a certain degree of adaptability and detection capabilities, which are partly derived from the goal-based framework proposed in this paper. This framework provides a foundational guideline for OWPTP implementation, underscoring the critical roles of goal prediction and epistemic uncertainty. Meanwhile, GMPDR's sparse representative replay strategies and motion pattern clustering are specifically designed for trajectory prediction tasks, which highlight the importance of transitioning from concrete instances to abstract patterns. Ultimately, these features demonstrate that GMPDR offers superior adaptability compared to other open-world learning methods when applied to complex and realistic OWPTP paradigm.

## G.5 Clustering and Embedding Visualization

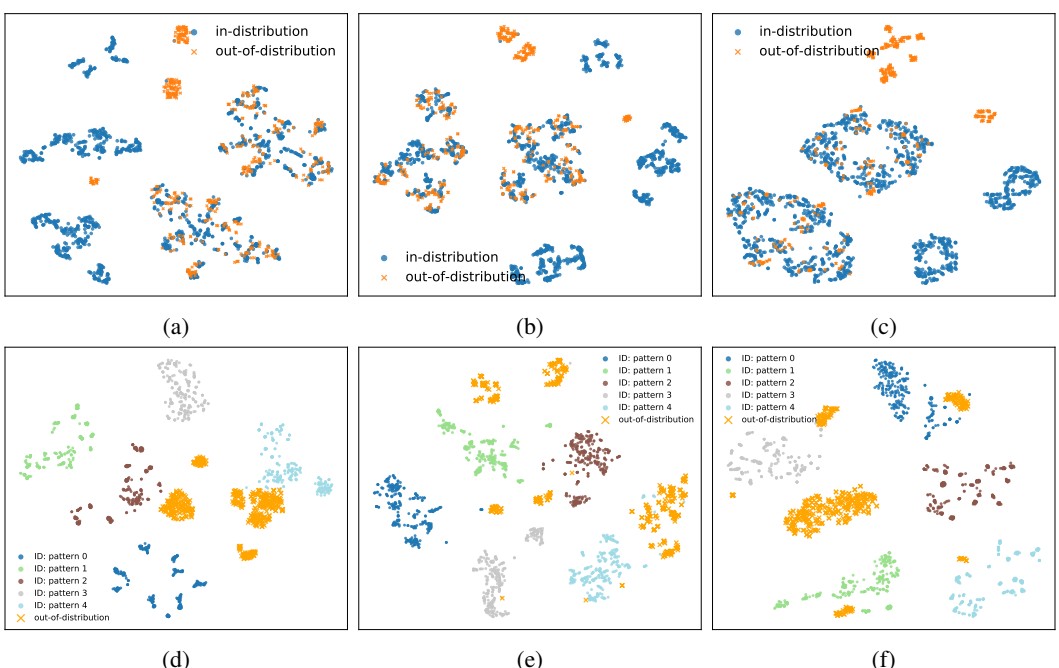

(a)  (b)  (c)

(d)  (e)  (f)

Figure 10: Comparing the embedding spaces of trajectory instances, each subfigure visualizes the embeddings of ID samples after one accommodation process and those of OOD samples during the subsequent detection process. (a), (b), (c) depict the embedding space of SHELS, while (d), (e), (f) illustrate the embedding space of the proposed GMPDR.

To further analyze the effectiveness of GMPDR, we visualize the clustering and embedding spaces generated on the SDD task, as shown in Fig. 10. The subfigures labeled "(d), (e), (f)" present the results of GMPDR. It can be observed that after an accommodation process, GMPDR successfully aggregates ID features into clearly distinguishable clusters through the MPDC module's clustering capability, thereby establishing corresponding clustering bases. As a result, a favorable instance embedding space is formed, as indicated by the distinct ID patterns in the subfigures, which facilitates the selection of representative replay samples. Subsequently, when novel motion pattern instances arrive, GMPDR maps the OOD samples into the embedding space, achieving a clear separation between ID and OOD samples, as illustrated by the orange-labeled examples in the subfigures. This process ultimately contributes to robust detection performance.

Comparing this with the SHELS model ((a), (b), (c)), the limitation of SHELS becomes evident in its inability to identify and group distinct motion patterns. As a result, all ID samples are uniformly treated and assigned to a single central vector. At the same time, the embeddings of OOD instances with novel trajectories partially overlap with those of ID trajectories. These issues jointly lead to less-than-optimal detection performance. From the visualization results, we observe that the ID embeddings generated by SHELS still exhibit a certain spatial structure, suggesting the presence of underlying motion patterns. Intuitively, one might consider applying clustering techniques, such as the K-means algorithm, to further segment the ID samples. However, this approach essentially aligns with the core idea proposed in this work: mapping specific trajectory instances to abstract motion patterns through clustering.

## H SUPPLEMENTARY ALGORITHM ANALYSIS

When evaluating the conditions for samples belonging to the ID category, we employ the dual OOD detection criteria. In the first criterion, we apply 8 augmentations to each sample and require that more than half of these augmented versions be assigned to the same cluster. In the second criterion, we assess whether the distance between each sample and its corresponding prototype is greater than the OOD threshold $\gamma_n \in \Gamma$, determined by the p-quantile.

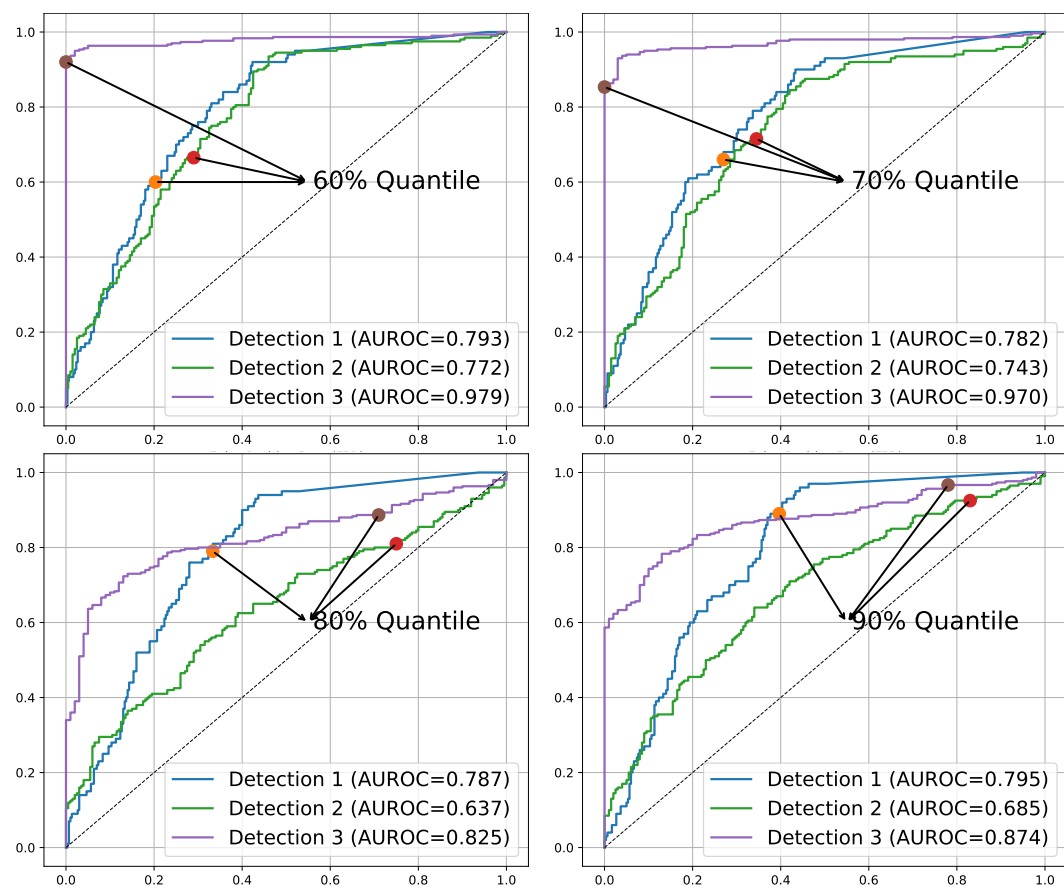

Figure 11: ROC curves and p-quantile positions for detection using multiple thresholds for calibration. The first criterion requires that more than 50% of the augmented versions belong to the same cluster.

To begin with, we focus on the influence of the p-quantile. However, since GMPDR utilises a multi-threshold OOD detection approach, where each pattern corresponds to a distinct OOD threshold $\gamma_n$, it is not feasible to directly construct a receiver operating characteristic (ROC) curve or mark the positions of different p-quantiles. To address this issue, we adopt a modified approach. Specifically, we select OOD thresholds $\Gamma$ corresponding to different p-quantiles and then standardise the distances computed after classifying each sample using the associated $\gamma_n$. This standardisation aligns all thresholds to a common scale, thereby enabling the construction of a calibrated ROC curve. Fig. 11 presents the ROC curves and the corresponding positions of $\Gamma$ selected based on different p-quantile for the SDD task. The three curves represent the detection performance across three instances of novelty introduction. Regardless of the p-quantile selected, the area under the ROC curve (AUROC) remains high, demonstrating the effectiveness of MPDC for OOD detection. Although GMPDR can determine an optimal p-quantile by analysing the ROC curve on the validation set, we recommend the use of the 70th percentile for generalization purposes. This choice achieves a higher AUROC while positioning $\Gamma$ in the upper-left region of the curve, corresponding to a higher true positive rate and lower false positive rate.

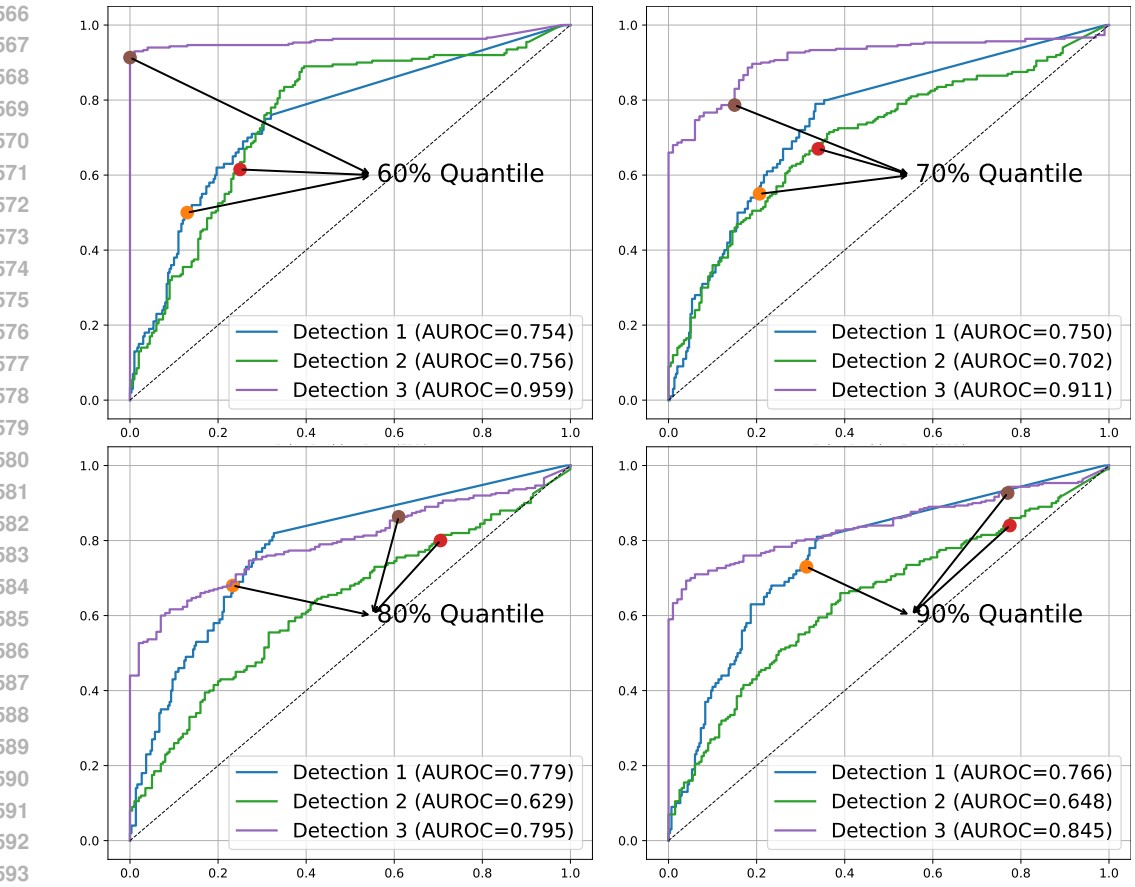

Figure 12: ROC curves and p-quantile positions for detection using multiple thresholds for calibration. The first criterion requires that more than 87.5% of the augmented versions belong to the same cluster.

We further tighten the first criterion by requiring that at least 7 out of 8 augmentations in each sample belong to the same cluster. Fig. 12 presents the calibrated ROC curves for different p-quantiles under this stricter condition. It can be seen that the 70th percentile remains an appropriate choice. Compared to the previously more lenient first criterion, the stricter criterion yields a slightly lower AUROC, as some ID samples are now misclassified as OOD under the stricter dual detection criteria. However, this stricter criterion achieves a lower false positive rate, as evidenced by the leftward shift in the p-quantile positions.

# I    THE USE OF LARGE LANGUAGE MODELS

We use an LLM solely as a general-purpose writing assistant for grammar and style polishing. All technical content is created and verified by the authors, who take full responsibility for the manuscript.

