# OpenReview forum: "Open-World Pedestrian Trajectory Prediction"
_ICLR.cc/2026/Conference — ICLR 2026 Conference Withdrawn Submission_

### Official Review · Reviewer_6V7V · 2025-10-28

**Soundness:** 2
**Presentation:** 1
**Contribution:** 2
**Rating:** 2
**Confidence:** 3

**Summary:**

This paper proposes a framework called **Goal-based Motion Pattern Detection and Replay (GMPDR)** for the newly defined task of **Open-World Pedestrian Trajectory Prediction (OWPTP)**. The method aims to detect novel motion patterns (OOD detection) and adapt to them (continual learning) by modeling epistemic uncertainty in goal prediction. The proposed GMPDR can be integrated into existing trajectory prediction methods to extend them to task of OWPTP.  Experiments on SDD and ETH/UCY datasets are provided to demonstrate effectiveness.

**Strengths:**

1. The paper attempts to address a challenging task, open-world trajectory prediction, by unifying novelty detection and continual learning.
2. The experiments cover several baselines and datasets, with many qualitative results and ablation studies.

**Weaknesses:**

1. The writing of the paper could be improved. The paper is difficult to follow due to unclear phrasing, ambiguous explanations, and confusing sentence structures. Many paragraphs lack logical flow, and transitions between paragraphs are abrupt.
2. The framework design is unclear. The role and necessity of components such as "LoRA adaptation" are insufficiently explained. It is hard to understand how each part contributes to the claimed detection and replay selection.
3. The formulations are not well explained. Several formulations (e.g., Eq. 3–8) are presented without clear explanation, making it difficult to understand the underlying logic and intuition.
4. The figures are unclear. For example, Figure 3 is too crowded and does not effectively illustrate the overall structure of the proposed method.

**Questions:**

1. How are the datasets $X^a$ and $X^b$ generated? The authors should clarify this process, as it is central to the method.
2. Why is LoRA necessary in the MPDC module? Its function and necessity should be better explained.
3. Why does the SDD dataset comprise four pattern sets and the ETH/UCY dataset comprise three pattern sets? How sensitive are the results to the choice of the number of pattern sets?
4. Could the authors provide experiments of the framework using simpler clustering algorithms such as *K-means* to validate the design choice?

---

### Official Review · Reviewer_gVKA · 2025-10-31

**Soundness:** 3
**Presentation:** 3
**Contribution:** 3
**Rating:** 4
**Confidence:** 5

**Summary:**

This paper studies pedestrian trajectory prediction in open-world settings, where models detect motion patterns that were not seen during offline training. It argues that the main source of motion pattern discriminability is uncertainty and forgetfulness about pedestrians’ goals. To address this, the authors propose a framework that first predicts a distribution over goals and then maps trajectories to motion patterns at the instance level. A key component, the Motion Pattern Dual Contrast (MPDC) module, performs contrastive clustering and builds hyperspherical embeddings to support per-pattern OOD detection. Experiments reported by the authors show that GMPDR adapts to novelty and reduces forgetting.

**Strengths:**

- The paper is clearly written and well organized.

- The introduction effectively sets up the problem, motivates the task, and presents the proposed solution and contributions.

- The method is described clearly, and the figures are informative and well designed.

- The appendix provides substantial implementation detail, which supports reproducibility, and the authors also make their code available.

**Weaknesses:**

- Test-time latency is not reported, which makes it difficult to assess the practicality of the method in real-time or multi-agent scenarios.

- There is no ablation study isolating the contribution of individual components (e.g., the different loss terms).

**Questions:**

- ETH/UCY has four unique scenes (ETH, Hotel, Univ, Zara1/2). Why is the initial number of patterns 3 for this dataset? Shouldn't it be four?

- Cross-dataset generalization is not discussed: Can a model trained on ETH/UCY transfer to a substantially different dataset such as SDD?

- After inference for each dataset, how many motion patterns are added?

- During inference, if a sample is deemed OOD, and after facing more OOD samples, the model gets trained on the new samples. Does that mean that gradually, the model is being trained on (some of) the test samples? If so, shouldn't comparison be with other models/baslines that are also trained with the same test samples?

- What is the ratio of the OOD samples in each dataset?

---

### Official Review · Reviewer_eDc3 · 2025-11-01

**Soundness:** 3
**Presentation:** 2
**Contribution:** 3
**Rating:** 6
**Confidence:** 3

**Summary:**

This paper first proposes the Open-World Pedestrian Trajectory Prediction (OWPTP), aiming to continuously detect and adapt to novel motion patterns in dynamic environments. OWPTP decomposes trajectory uncertainty into epistemic and aleatoric components. Then the authors propose the Goal-based Motion Pattern Detection and Replay (GMPDR) framework, which models epistemic uncertainty to identify motion-pattern-level novelty via a dual-contrastive module and mitigates forgetting through sparse representative replay. Experiments on multiple datasets demonstrate that GMPDR outperforms existing baselines.

**Strengths:**

+ The paper introduces the Open-World Pedestrian Trajectory Prediction (OWPTP) paradigm, which is an important and previously underexplored challenge in trajectory forecasting. This new formulation aligns well with realistic deployment scenarios and is conceptually well-motivated.
+ The experiments cover multiple datasets, metrics, baselines, and ablations, demonstrating consistent improvements in both detection and continual learning performance.
+ The proposed Motion Pattern Dual Contrast (MPDC) module provides a principled mechanism to form explicit pattern-level representation spaces, enabling interpretable motion pattern boundaries and more robust OOD decision-making.

**Weaknesses:**

- Fig. 2(b) is confusing. The meaning of the vertical axis is not explained. And the corresponding text is also confusing (line 170-174). The authors should further explain this experiment and the figure.
- The definition of f_i, q_theta, mu_j, and the feature set F in Sec 4.2.1 is unclear.
- Others in questions.

**Questions:**

1. Fig 1 is confusing. What is the relationship between the “Novel Trajectory”, “Learned Trajectory”, and “Ground Truth”? And the “Pre-predict”, “Post-predict”, “After-forget” are confusing too.
2. Line 63-65: “Prior research indicates that features encoded from trajectories capture their pattern information. However, this information remains implicit and inadequate.” Why is this information inadequate? Is there any experiment to support this conclusion?
3. According to Sec 3.1, when the model detect a new motion pattern, it will not predict but will train on this pattern. Then, the problem is how to divide the training dataset and the test dataset? If the test dataset contains motion patterns that are not included in the training dataset, will the model train on these patterns too?
4. Why use LoRA instead of training a new encoder?
5. Why use softmax instead of a clustering algorithm like KMeans to implement clustering?

---

### Official Review · Reviewer_aULn · 2025-11-03

**Soundness:** 3
**Presentation:** 3
**Contribution:** 3
**Rating:** 6
**Confidence:** 2

**Summary:**

This paper proposes Open-World Pedestrian Trajectory Prediction (OWPTP), a new paradigm that enables trajectory models to continuously learn and adapt to unseen motion patterns in dynamic, real-world environments. The authors introduce the GMPDR (Goal-based Motion Pattern Detection and Replay) framework, which detects novel motion behaviors through goal-based uncertainty modeling and adapts via dual-contrast representation learning and sparse replay. Experiments on multiple datasets demonstrate strong performance in both novelty detection and continual adaptation.

**Strengths:**

The motivation is clear and meaningful, addressing the real challenge of adapting trajectory models to unseen motion patterns in open-world settings.
The paper is well organized and logically presented, making the framework and methodology easy to follow.
The experiments are kind of extensive.

**Weaknesses:**

The scalability and real-time applicability of the continual learning process are unclear, frequent replay and feature clustering may become costly in large-scale or streaming environments. Although the paper claims efficiency through sparse representative replay, no explicit experiments or runtime analyses are provided to substantiate this claim.

The framework is evaluated mainly on pedestrian datasets. It would be interesting to see whether the proposed approach could generalize or be extended to other domains, such as vehicles, cyclists, or mixed-traffic scenarios, where motion dynamics and interaction patterns differ, more like an open-world scenario.

The paper would benefit from an explicit limitations discussion, acknowledging current constraints (e.g., dependency on goal hypotheses, computational overhead, or dataset diversity) and outlining how future work could be organized.

**Questions:**

pls refer to the Weaknesses

---

### Note · Authors · 2025-11-13

**Comment:**

We sincerely thank the reviewers for their thoughtful assessments and constructive suggestions. This paper studies Open-World Pedestrian Trajectory Prediction (OWPTP) and proposes GMPDR, a unified framework that models goal-based uncertainty and, via the MPDC module, performs pattern-level OOD detection and sparse representative replay. We observed stable and autonomous detect-accommodate cycles under multiple novelty introductions, operating with a very small replay ratio and a modest memory footprint. We appreciate the reviewers’ positive feedback on the motivation, method and experiments, and we acknowledge that several explanations were not sufficiently clear, which hindered communication of key ideas. To better address the feedback, we are withdrawing the submission for revision.

**Withdrawal Confirmation:**

I have read and agree with the venue's withdrawal policy on behalf of myself and my co-authors.